# Collab: Controlled Decoding using Mixture of Agents for LLM Alignment

**Souradip Chakraborty**[1,2] *   **Sujay Bhatt**[1]   **Udari Madhushani Sehwag**[1]   **Alec Koppel**[1]
**Soumya Suvra Ghosal**[2]   **Jiahao Qiu**[3]   **Mengdi Wang**[3]   **Dinesh Manocha**[2]
**Furong Huang**[2]   **Sumitra Ganesh**[1]
[1]JPMorgan AI Research   [2]University of Maryland, College Park   [3]Princeton University

## Abstract

Alignment of Large Language models (LLMs) is crucial for safe and trustworthy deployment in applications. Reinforcement learning from human feedback (RLHF) has emerged as an effective technique to align LLMs to human preferences, and broader utilities, but it requires updating billions of model parameters which is computationally expensive. Controlled Decoding, by contrast, provides a mechanism for aligning a model at inference time without retraining. However, single-agent decoding approaches often struggle to adapt to diverse tasks due to the complexity and variability inherent in these tasks. To strengthen the test-time performance w.r.t the target task, we propose a mixture of agents-based decoding strategies leveraging the existing off-the-shelf aligned LLM policies. Treating each prior policy as an agent in the spirit of mixture of agent collaboration, we develop a decoding method that allows for inference-time alignment through a token-level selection strategy among multiple agents. For each token, the most suitable LLM is dynamically chosen from a pool of models based on a long-term utility metric. This policy-switching mechanism ensures optimal model selection at each step, enabling efficient collaboration and alignment among LLMs during decoding. Theoretical analysis of our proposed algorithm establishes optimal performance with respect to the target task represented via a target reward, for the given off-the-shelf models. We conduct comprehensive empirical evaluations with open-source aligned models on diverse tasks and preferences, which demonstrates the merits of this approach over single-agent decoding baselines. Notably, Collab surpasses the current SoTA decoding strategy, achieving an improvement of up to 1.56x in average reward and 71.89% in GPT-4 based win-tie rate.

## 1 Introduction

Large language models (and generative models) excel at generating coherent and realistic text, but many text generation tasks require outputs that not only preserve fluency but also require satisfying constraints, such as factual accuracy (Wang et al., 2024), knowledge grounding (Liang et al., 2024), adherence to safety guidelines (Dong et al., 2024; Xie et al., 2024), or task and domain-specific objectives (Jeong, 2024). Ensuring personalized or task-specific alignment in LLMs requires fine-tuning them for specialized objectives. Alignment approaches like reinforcement learning from human feedback (RLHF) (Ouyang et al., 2022; Stiennon et al., 2022; Ziegler et al., 2020; Yuan et al., 2023; Chakraborty et al., 2024c) have shown efficiency in aligning generative models with human and task-specific preferences. However, fine-tuning billions of parameters with RLHF is computationally intensive and becomes impractical in the context of personalized or highly specialized alignment. Controlled Decoding (Mudgal et al., 2024) aims to address this challenge by providing a training-free inference-time framework that allows models to be aligned with target preferences and tasks without requiring the retraining of billions of model parameters. Recent works (Mudgal et al., 2024; Khanov et al., 2024; Chakraborty et al., 2024b) have demonstrated that inference-time alignment can significantly enhance the ability of large language models to meet task-specific and personalized

---

*Work done as a part of an internship at JPMorgan AI Research. Correspondence: Souradip Chakraborty <schakra3@umd.edu>, Alec Koppel <alec.koppel@jpmchase.com>.

requirements, even improving the fine-tuning based algorithms (Khanov et al., 2024; Mudgal et al., 2024; Chakraborty et al., 2024b). These methods, while effective for single-agent decoding, are limited when it comes to handling diverse or conflicting task requirements, as they struggle to generalize across tasks that demand fundamentally diverse or specialized capabilities.

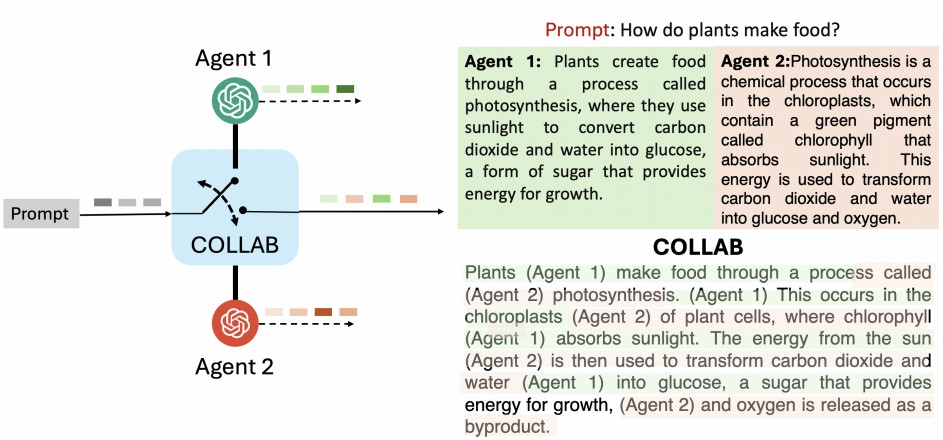

Figure 1: The figure illustrates the optimal coordination between agents for response generation via switching, where Agent1 is a ChatAgent and Agent2 is a Chemical-Expert. In this collaborative response, the agents are switching smoothly at the word and phrase level to deliver a more detailed and complete response than they could individually. The switching demonstrates how both agents complement each other in explaining the complex process.

**Limitation with Single agent decoding:** The performance of single-agent decoding approaches often struggles to adapt to diverse tasks due to the complexity and variability inherent in these tasks and also due to their over-reliance on the training task distribution. Specialized tasks may demand conflicting capabilities: for *example, one task might require fact-based grounding, such as retrieval or summarization, where accuracy and precision are critical, while another may involve creative outputs like writing poetry or fiction, where imagination and stylistic freedom are more important* These conflicting demands make it difficult for a single-agent decoding method to generalize effectively and meet the distinct requirements of both fact-driven and creative tasks simultaneously.

**Challenges in Multiagent Decoding for Alignment**: A straightforward solution to the above challenges is to leverage multiple existing off-the-shelf LLMs, each specializing in different tasks and domains, to collaborate efficiently during inference. However, a key challenge lies in efficiently combining these LLMs in a tuning-free manner without requiring retraining for each new task. Current methods that attempt to integrate multiple LLMs often rely on weak supervision or explicit formulas for mixing model outputs (Shen et al., 2024; Jin et al., 2024; Yang & Klein, 2021; Liu et al., 2024; Mialon et al., 2023), but these approaches lack the flexibility to adapt dynamically during token-level generation. Additionally, most prior methods rely on expert supervision to guide the combination of logits or determine when to leverage other LLMs, which can be restrictive. Moreover, many existing approaches require some form of training to integrate multiple LLMs, a process that can be expensive or infeasible when access is limited. To the best of our knowledge, none of the prior decoding alignment methods have thoroughly explored the optimal selection of specialized agents during inference, which requires defining a principal metric for selection. Thus we ask the question.

Recent methods for integrating multiple LLMs often rely on weak supervision or fixed formulas to mix outputs, lacking flexibility for dynamic token-level generation. They typically require expert guidance for combining logits or deciding when to leverage other models, which can be restrictive.

*How can existing off-the-shelf models be optimally combined during inference to target specific tasks without retraining, allowing for dynamic collaboration and adaptation at the token level?*

**Mixture of Agents based Controlled Decoding**: Hence, in this work, we leverage and combine the strengths of multiple off-the-shelf LLM models, each aligned with specialized or diverse tasks, to enable decoding without retraining, in the spirit of a mixture of agents approach. More specifically, each agent is aligned with specialized preferences through distinct reward functions (Ouyang et al.,

2022; Woźniak et al., 2024). We propose a mixture-of-agents-based controlled decoding strategy that aligns the final response to a new task without retraining via aggregating the next-token prediction distributions from each agent during the response generation phase. However, to generate the response optimally by combining the LLMs, it is crucial to address the question of how to design an effective aggregation strategy (decoding policy) that combines the outputs from the mixture of agents.

**Switching between Mixture of Agents with Q-function** : To address the question of designing a suitable decoding policy, we identify that optimal decoding is fundamentally guided by the long-term utility with respect to a semantic reward, often characterized by the Q-function (Mudgal et al., 2024; Chakraborty et al., 2024b). This allows us to define an optimal policy by framing the decoding process through the lens of a KL-regularized reinforcement learning problem. This framework is specifically crafted to solve the alignment problem by selecting the policy for each token during decoding to be optimal with respect to the aforementioned implicit Q function. We introduce COLLAB Controlled decoding via Mixture of Agents as a potential solution for test-time collaborative alignment, with an implicit Q-function-serving as the guiding metric for decoding. Our proposed method is grounded in a policy-switching mechanism, where the implicit Q-function is used to optimally select the aligned model at each time step during decoding. We define the notion of the implicit Q-function within the mixture of agents decoding algorithm and demonstrate its effectiveness as an optimal alignment metric, both theoretically and empirically. We show that the implicit Q-function provides a principled approach for selecting the model that best aligns with the target task achieving the optimal performance possible under the given scenario. We summarize our key contributions as follows:

- **Mixture of Agents Decoding Strategy for Alignment** We introduce a novel mixture-of-agents decoding strategy that leverages specialized off-the-shelf LLM agents, each aligned with a specific task or reward function. Our method optimally switches between these agents during decoding to achieve alignment with the target reward function, without requiring retraining.

- **Collaborative Metric for Mixture of Agent Decoding**: We propose the concept of the implicit Q-function as a guided metric to combine the mixture of agents for decoding. This metric enables an appropriaye combination of the off-the-shelf LLM agents during inference w.r.t the target objective.

- **Theoretical Characterization of COLLAB**: We provide a precise theoretical characterization of our proposed approach via analyzing the sub-optimality gap w.r.t to the target reward. We characterize the sub-optimality of our approach w.r.t the true Q-function by upper-bounding the performance gap through the reward difference between the target and best model for each token, along with the KL divergence from the reference policy.

- **Experimental Evaluations**: Through extensive empirical evaluations, we demonstrate that our proposed decoding approach significantly outperforms state-of-the-art single-agent decoding baselines in task alignment performance, particularly in scenarios involving diverse or conflicting requirements. We also conducted comprehensive evaluations using various metrics—including average reward, GPT-4 win rate, coherence, and diversity—to showcase the superiority of our approach. Additionally, we performed ablation studies to show that diversity among agents enhances collaborative performance in the mixture.

## 2  RELATED WORK

Alignment via fine-tuning with reinforcement learning from human feedback (RLHF) has emerged as a key paradigm for aligning foundation models (Ouyang et al., 2022; Chakraborty et al., 2024a; Rafailov et al., 2023; Chakraborty et al., 2024c; Chen et al., 2024). RLHF typically operates in two phases: first, a reward model is trained on human feedback, and then a policy is fine-tuned with reinforcement learning (PPO (Schulman et al., 2017)) using the trained reward model (Ouyang et al., 2022; Stiennon et al., 2022; Ziegler et al., 2020; Yuan et al., 2023; Go et al., 2023; Vamplew et al., 2018; 2008). Direct preference optimization seeks to stabilize the training of preference alignment through reductions to supervised learning training (Rafailov et al., 2023). Although these training methods have proven effective in aligning generative models, they remain computationally demanding and assume white-box access to the model parameters which is not true in many industry applications.

On the other hand, decoding-based methods (Mudgal et al., 2024; Chakraborty et al., 2024b) have emerged as an alternate way of alignment without fine-tuning the model parameters. Decoding operates by altering the distribution of the generated response to align to the target preference directly

without updating the parameters of the LLM. The work by (Mudgal et al., 2024) is one of the first to integrate the alignment procedure directly into the decoding process, where they propose adjusting the generation probabilities at each decoding step based on feedback from a reward model. (Huang et al., 2024) redefined the text-generation process as a search problem, with LLMs acting as search agents and they employ a heuristic-guided search mechanism to generate responses based on a given prompt.The most recent research around Controlled and Principled decoding (CD, TQ*) formulates the decoding problem as a KL regularized reinforcement learning problem and obtains a closed-form solution with an estimate of $Q^*$ for decoding. However, the majority of these prior approaches are inherently single-agent alignment strategies and lack the ability to leverage the strengths of diverse agents in limiting their effectiveness in complex, multi-faceted tasks. Recent methods attempt to integrate multiple LLMs either rely on weak supervision or explicit formulas for combining model responses (Shen et al., 2024; Jin et al., 2024; Yang & Klein, 2021; Liu et al., 2024; Mialon et al., 2023), which might be restrictive.

## 3 PROBLEM FORMULATION: COLLABORATIVE MULTIAGENT DECODING

### 3.1 PROBLEM DEFINITION

**Task and Preference Representation with Target Reward Function**: To address the problem of optimal response generation, we first define the framework around a specialized target score or reward function $r_{\text{target}}$ which serves as the key objective guiding response generation. The reward function $r_{\text{target}}$ encapsulates a wide range of tasks and objectives, thereby allowing for flexibility and adaptability. These can include: *1. Personalized alignment:* $r_{\text{target}}$ can be tailored to individual preferences or specialized tasks, ensuring personalized responses that align with user-specific goals, *2. Rules-based approaches:* Various rules or heuristics used in specialized contexts can be represented as constraints or structured forms of the reward function, making it possible to enforce domain-specific guidelines, *3. Supervised signals:* Any supervised learning signals or labels provided during training can be mapped as rewards, allowing the learning system to directly optimize for them, *4. Open-source reward models:* Many publicly available reward models that have been fine-tuned for specific objectives can be expressed under $r_{\text{target}}$ broadening the applicability across multiple domains. By casting these diverse objectives within the framework of a reward function, we ensure that the system is capable of handling a variety of alignment and optimization tasks through the target reward function.

**Response Generation Under Specialized Reward Functions:** Given a target reward function, the challenge lies in determining the optimal way to generate a response from the LLM agent and it becomes especially difficult when the target reward significantly deviates from the reward associated with the aligned policy. Single-agent systems often struggle with adapting to diverse tasks, primarily due to: **(1) Task Complexity and Variability:** Tasks differ significantly in their structure and requirements; for example, one task might require generating long-form responses, while another might favor brevity. Similarly, some tasks might demand reasoning suited to an expert audience, such as PhD-level logic, while others might need responses tailored to general understanding (Jang et al., 2023; Woźniak et al., 2024) **(2) Training Task Over-reliance:** Single agent alignment methods often overfit to the distribution of training tasks or reward function, leading to reward overfitting or overoptimization (Gao et al., 2023; 2022; Zhu et al., 2024) resulting in limited generalization to target reward and tasks.

**Mixtue of Agents for Alignment**: However, we note that there are already existing available off-the shelf LLM policies aligned to a diverse of set of tasks (Lambert et al., 2024; Jang et al., 2023; Woźniak et al., 2024). Hence, to address these limitations, we leverage the set of already available specialized LLM policies represented as $\Pi = \{\pi_1, \pi_2 \cdots \pi_k\}$ aligned to a set of tasks (or preferences) as defined by specific reward functions belonging to set $\mathcal{R} = \{r_1, r_2 \cdots r_k\}$. These policies can collectively handle a diverse range of tasks, allowing for more flexible and effective adaptation.

**Challenge in Multiagent Decoding for Alignment**: The challenge remains: the reward functions $\{r_1, r_2 \cdots r_k\}$ are latent and unobservable, often because they are proprietary or embedded within the model's training data, which may not be available for external analysis. Thus, transferring efficiently to the target reward involves adapting the response generation process based on the information from these pre-trained and given input models without explicit access to their internal reward functions leading to the key questions in Multi-agent decoding for alignment.

**Key Questions in Multi-Agent Decoding:** One of the key challenges when decoding with multiple agents is determining when to switch between models and the second challenge lies in identifying the appropriate metric to guide this decision.

**Formalizing the Problem with Markov Decision Processes**: We provide a formal answer to these questions by defining the response generation phase with an appropriate Markov Decision Process with a specific reward function, and its associated optimality criteria. Doing so is the focus of the following subsection.

### 3.2 Token-level Markov Decision Process

We begin by formulating the decoding problem as a KL regularized reinforcement learning problem (Mudgal et al., 2024; Chakraborty et al., 2024b) with the token-level MDP $\mathcal{M} := \{\mathcal{S}, \mathcal{A}, P, R\}$ where the state-space $\mathcal{S}$ represents the concatenated sequence of tokens and the action space $\mathcal{A}$ representing the space of the next token i.e vocabulary $\mathcal{V}$. Given a state $\mathbf{s}_t = [\mathbf{x}, \mathbf{y}_{<t}] \in \mathcal{S}$, which is a sequence of tokens containing the prompt/query $\mathbf{x} := \{x_1, x_2, \cdots, x_N\}$ appended with the $t$ tokens $\mathbf{y}_{<\mathbf{t}} := \{y_0, y_1, \cdots, y_{t-1}\}$ generated so far, an LLM is a policy $\pi$ that samples actions as next sampled tokens the action (i.e., the next token) $a_t = y_t$ via sampling from the token-level decoding policy $y_t \sim \pi(\cdot \mid \mathbf{s}_t)$. The transition $P$ to the next state $\mathbf{s}_{t+1}$ is deterministic: $\mathbf{s}_{t+1} = [\mathbf{x}, \mathbf{y}_{<t}, y_t]$, the concatenation of the current state and action. We denote the trajectory level probability by $\rho_\pi(z|\mathbf{x}) = \prod_{t=1}^T \pi(y_t|[\mathbf{x}, \mathbf{y}_{<t}])$ and $\tau \sim \rho_\pi(\cdot|\mathbf{x})$ represents a sampled response/trajectory which is a concatenation of tokens, and $z$ is an arbitrary token.

The reward $r(\mathbf{s}, \mathbf{a}) : \mathcal{S} \times \mathcal{A} \to \mathbb{R}$. To be specific, the action-value function associated with the reward is then defined for a length $L$ as

$$Q^\pi(\mathbf{s}, z) = \mathbb{E}_\pi \Big[ \sum_{t=0}^{L-1} r(s_t, z_t) \Big| s_0 = \mathbf{s}, z_0 = z; z_t \sim \pi(\cdot|[\mathbf{x}, \mathbf{y}_{<\mathbf{t}}]) \Big]$$
$$:= \mathbb{E}_{\tau' \sim \rho_\pi(\cdot|\mathbf{s}, z)} \left[ r([\mathbf{s}, z], \tau') \mid \mathbf{s}, z \right], \tag{1}$$

where $\mathbf{s}$ is the state and $z \in \mathcal{V}$ the action for the state and $\tau'$ is sampled from the trajectory distribution $\rho_\pi(\cdot|\mathbf{s}, z)$ induced by the policy $\pi(\mathbf{s}, z)$.

### 3.3 Decoding Process and Controlled Decoding

In this section, we introduce the decoding process where in we have a reference policy $\pi_{\text{ref}}$ which takes as an input the prompt $x \in \mathcal{V}^N$ (a string of N tokens) and generates a response $\mathbf{y} = [y_0, y_1, \cdots, \text{EOS}]$ token by token with a probability of each token $t$ is given by $\pi(\cdot|[\mathbf{x}, \mathbf{y}_{\leq\mathbf{t}}])$. With $z \in \mathcal{V}$ as a token from vocabulary set $\mathcal{V}$, the objective of LLM alignment via decoding is formally defined by solving the KL regularized MDP $\mathcal{M}$ as

$$\pi^*(\cdot|\mathbf{s}_\mathbf{t}) := \arg\max_\pi \mathbb{E}_{z \sim \pi(\cdot|\mathbf{s}_\mathbf{t})} \left[Q^*(\mathbf{s}_\mathbf{t}, z)\right] - \alpha \mathbb{D}_{\text{KL}} \left[\pi(\cdot|\mathbf{s}_\mathbf{t})||\pi_{\text{ref}}(\cdot|\mathbf{s}_\mathbf{t})\right], \tag{2}$$

where we note that $\mathbf{s}_\mathbf{t} = [\mathbf{x}, \mathbf{y}_{\leq\mathbf{t}}]$, and $Q^*(\mathbf{s}_\mathbf{t}, z)$ denotes the optimal state-action value function for the token-level MDP $\mathcal{M}$ defined in the previous section. The KL constraint ensures closeness to the reference policy $\pi_{\text{ref}}(\cdot|\mathbf{s}_t)$, with hyperparameter $\alpha > 0$. The closed-form solution of the problem as

$$\pi^*(z|\mathbf{s}_\mathbf{t}) = \pi_{\text{ref}}(z|\mathbf{s}_\mathbf{t}) \frac{\exp\left(\frac{1}{\alpha}Q^*(\mathbf{s}_\mathbf{t}, z)\right)}{C_\alpha(\mathbf{s}_t)}, \tag{3}$$

where $C_\alpha(\mathbf{s}_t) := \sum_z \pi_{\text{ref}}(z|\mathbf{s}_\mathbf{t}) \exp\left(\alpha Q^*(\mathbf{s}_\mathbf{t}, z)\right)$ is the normalizing constant for state $\mathbf{s}_\mathbf{t}$ (Mudgal et al., 2024). However, it is important to note that $Q^*(\mathbf{s}_\mathbf{t}, z)$ is the optimal state-action value function and is unavailable in practice. Recent works by (Khanov et al., 2024; Chakraborty et al., 2024b; Mudgal et al., 2024) design approximate methods to estimate this $Q$ function and thus adjust the probability of the reference policy with exponential of $Q^*$ as in equation 2 to obtain the final decoding policy. Recent works have also characterized the sub-optimality of decoding under such approximations (Chakraborty et al., 2024b). However, the entire decoding process is primarily defined for single-agent systems, and in particular, a single training distribution and aligned policy. Extending this problem class to multiple agents is the goal of this work.

### 3.4 Proposed method: Mixture of Agent-based Controlled Decoding for Alignment

With the problem of sampling from a single LLM agent being well-defined, we now turn to the challenge of decoding from multiple off-the-shelf LLM policies. We introduce our proposed approach of the mixture of agent-based controlled decoding by leveraging existing off-the-shelf LLMs for better generalization to target reward function. The objective is to efficiently align to the target reward function $r_{\text{target}}$ with the off-the-shelf LLM policies $\Pi = \{\pi_1, \pi_2 \cdots \pi_k\}$ to hidden reward functions $\mathcal{R} = \{r_1, r_2 \cdots r_k\}$. If we had access to the true aligned policy $\pi^*_{\text{target}}$ for the target reward function $r_{\text{target}}$, then we could have followed similar steps from prior works on decoding (Chakraborty et al., 2024b; Mudgal et al., 2024), and decode using the final equation 2, where we can estimate the $Q^*$ as

$$Q^*_{\text{target}}(\mathbf{s_t}, z) = \mathbb{E}_{\tau \sim \rho_{\pi^*_{\text{target}}}(\cdot|\mathbf{s}_t, z)}\left[r_{\text{target}}([\mathbf{s_t}, z], \tau)|\mathbf{s_t}, z\right] \tag{4}$$

where $\rho_{\pi^*_{\text{target}}}$ is the corresponding trajectory level policy (joint-distribution) under the target reward function $r_{\text{target}}$, defined in Section 3.2. However, as previously mentioned, $\pi^*_{\text{target}}$ is unavailable for the target reward $r_{\text{target}}$. Thus our proposed approach focuses on designing a collaborative strategy to leverage existing off-the-shelf LLMs policies $\Pi = \{\pi_1, \pi_2 \cdots \pi_k\}$ to decode to the target reward $r_{\text{target}}$. Thus, one may not conduct a general policy search, but instead restrict focus to a feasible set of prior models $\Pi$, formalized as:

$$\pi^*(\cdot|\mathbf{s}_t) := \arg\max_{\pi \in \Pi} \mathbb{E}_{z \sim \pi(\cdot|\mathbf{s}_t)}\left[Q^*_{\text{target}}(\mathbf{s}_t, z)\right] - \alpha \mathbb{D}_{\text{KL}}\left[\pi(\cdot|\mathbf{s}_t)||\pi_{\text{ref}}(\cdot|\mathbf{s}_t)\right], \tag{5}$$

where, the KL-regularized RL problem for each token is defined over the constrained set $\pi \in \Pi$, $\pi_{\text{ref}}$ is the reference policy $Q^{\pi}_{\text{target}}(\mathbf{s}_t, z)$ is the long-term action-value function for that corresponding policy, given as

$$Q^{\pi_j}_{\text{target}}(\mathbf{s}_t, z) = \mathbb{E}_{\tau \sim \rho^{\pi_j}(\cdot|\mathbf{s}_t, z)}\left[r_{\text{target}}([\mathbf{s}_t, z], \tau)\right] \tag{6}$$

For simplicity of notations, we denote the objective for the $j^{\text{th}}$ agent as $J^{\pi_j}_{\text{target}}(\mathbf{s}_t, z) = Q^{\pi_j}_{\text{target}}(\mathbf{s}_t, z) - \alpha KL(\pi_j(\cdot|s), \pi_{\text{ref}}(\cdot|s))$. Thus, for each state $\mathbf{s}_t = [\mathbf{x}, \mathbf{y}_{<t}]$, where $\mathbf{x}$ is the prompt and $\mathbf{y}_{<t}$ is the sequence of tokens till time $t$, the optimal choice of the next token is given as $z^* := \arg\max_z \max_j J^{\pi_j}_{\text{target}}(\mathbf{s}_t, z)$. A detailed description of our policy switching algorithm is presented in algorithm 1 and discussed in the next section.

**Mixture of Agents-based Decoding via Switching:** Our proposed mixture of agents-based decoding approach induces a policy-switching mechanism among the individual LLM agents at each state $\mathbf{s}_t$, aimed at generating a final response aligned with the target reward $r_{\text{target}}$. For each state $\mathbf{s}_t = [x, y_{<t}]$ our approach identifies the optimal LLM agent for decoding based on the maximum value of the long-term metric $J^{\pi_j}_{\text{target}}(\mathbf{s}_t, z)$ which we denote as the implicit Q-function. Subsequently, it selects the token $y_t = z$ generated using that agent based on our algorithm's decoding policy as

$$\pi_{\text{alg}} \in \arg\max_z \max_j J^{\pi_j}_{\text{target}}(\mathbf{s}_t, z) \tag{7}$$

This token is then concatenated with the previously generated response, forming the next state $\mathbf{s}_{t+1} = \mathbf{s}_t \cup y_t$ which is subsequently used as a prompt for all agents for the next time steps. Thus, the mixture of agents implicitly collaborates, with each agent conditioned on prompts generated through a policy switching mechanism (see Sec.F). This dynamic interaction yields an improved response, better aligned with the target reward $r_{\text{target}}$. We present the detailed flow of our proposed approach in algorithm 1.

## 4 Theoretical Analysis of Mixture of Agents Decoding for Alignment

In this section, we analyze the sub-optimality gap of our proposed Collaborative mixture of agent decoding strategy w.r.t the target reward functions $r_{\text{target}}$. We quantify sub-optimality of a decoding method $\pi$ w.r.t the long-term action-value $Q$ function for the target reward.

$$\Delta(\pi) = Q^{\pi^*}_{\text{target}}(\mathbf{s}_t, z) - Q^{\pi}_{\text{target}}(\mathbf{s}_t, z) \tag{8}$$

where $\pi^*$ represents the optimal policy under the target reward function $r_{\text{target}}$ and $\pi$ is an arbitrary decoding policy induced. Subsequently we study equation 8 when decoding according to $\pi = \pi_{\text{alg}}$ as defined in Algorithm 1.

---

**Algorithm 1** Mixture of Agents based Controlled Decoding for LLM Alignment

---

**Input:** Set of LLM policies $\Pi = \{\pi_1, \pi_2 \cdots \pi_K\}$ aligned to a diverse set of latent reward functions, target reward function $r_{\text{target}}$, dataset of prompts $x \in D$, decoding parameter $\alpha$, vocabulary set $\mathcal{V}$

**for** $t = 0, \ldots, T$ **do**

    **Current state:** $\mathbf{s}_t = [\mathbf{x}, \mathbf{y}_{<\mathbf{t}}]$, with prompt $\mathbf{x}$ and prior token sequence $\mathbf{y}_{<\mathbf{t}} = [y_0, y_1, \cdots, y_{t-1}]$

    **for** $i = 1, \ldots, K$ **do**

        **Sample** top-p tokens using $i^{\text{th}}$ aligned policy $\pi_i$ and store as $\mathcal{V}_i = \{z_p^i : z_p^i \sim \pi_i(\cdot|\mathbf{s}_t)\}_{p=1}^p$

        **for** $z \in \mathcal{V}_i$ **do**

            **Evaluate** the implicit Q-function for the $i^{\text{th}}$ policy $J_{\text{target}}^{\pi_i}(\mathbf{s}_t, z)$ using equation 5

            **Estimate** the best token for the $i^{\text{th}}$ policy as $z_{\max}^i = \arg \max_{z \in \mathcal{V}_i} J_{\text{target}}^{\pi_i}(\mathbf{s}_t, z)$

    **Select** $z_{\max} = \arg \max_i J_{\text{target}}^{\pi_i}(\mathbf{s}_t, z_{\max}^i)$

    **Next Token:** $y_{t+1} \leftarrow z_{\max}$

    **Next State:** $\mathbf{s}_{\mathbf{t+1}} \leftarrow [\mathbf{s}_t, y_{t+1}]$

**Return:** $\mathbf{y}^* = [y_0^*, \ldots, y_T^*]$

---

> **Example of response generated using different strategies**
>
> **Prompt:**
> **Human:** 10+10+10-30 equals what
> **Assistant:** you're missing a positive sign... the result is positive.
> **Human:** No, it's a math equation. Let's try it another way. What is 10 plus 10 plus 10 minus 30?
> **Assistant:**
>
> ---
>
> **Agent-I:** 10 + 10 + 10 - 30 equals -10, because the positive numbers add up to 30, and then you're subtracting 30, which leaves you with the negative result of -10. If you meant to add the numbers before subtracting, then the result would be 30, as you have 30 after adding 10, 10, and 10, and before the subtraction of 30.
>
> ---
>
> **Agent-II:** 10 + 10 + 10 - 30 equals -10. Because when you add 10, 10, and 10, you get 30. Then, when you subtract 30, you get -10. It's like you're taking 30 away from a balance, and you end up in the negative.
>
> ---
>
> **BoN Sampling:** This question has three separate elements. 10+10+10 is 100, and -30 is -30. The entire question can be expressed as 100 - 30.
>
> ---
>
> **COLLAB (Ours):** 10 + 10 + 10 - 30 = 30 - 30 = 0, Explanation: To evaluate the expression, perform the addition and subtraction in order from left to right: 1. Add 10 and 10 to get 20: 10 + 10 = 20, 2. Add 10 to the previous sum: 20 + 10 = 30

**Theorem 1** (Sub-optimality Bound of Multi-Agent Decoding Algorithm). *Let $\Pi = \{\pi_1, \pi_2, \ldots, \pi_K\}$ be a set of pre-trained policies, each aligned to a latent reward function $r_j$, and $\pi_{alg}$ be the policy obtained by the multi-agent decoding strategy. Assume that the optimal policy for the target reward function $r^*$ is $\pi^*$. Then, the sub-optimality of the multi-agent decoding policy $\pi_{alg}$ with respect to the optimal policy $\pi^*$ is bounded by:*

$$\Delta(\pi_{alg}) \leq \min_{j \in K} \delta_{*j} + \alpha KL(\pi_j(\cdot|\mathbf{s}_t), \pi_{ref}(\cdot|\mathbf{s}_t))] + \beta KL(\rho^{\pi^*}(\cdot|\mathbf{s}_t), \rho_{ref}(\cdot|\mathbf{s}_t)) \tag{9}$$

*where $\delta_{*j} = \max_\tau |r_{target}([\mathbf{s}_t, z], \tau) - r_j([\mathbf{s}_t, z], \tau)|$, $\alpha, \beta > 0$ are regularization constants for the KL-divergence terms to ensure closeness to the reference policy at the token and trajectory level, $\delta_{*j}$ represents the difference between the target reward function and the reward function of the closest to the target. $KL(\pi(\cdot|s), \pi_{ref}(\cdot|s))$ represents the KL-divergence between policy $\pi$ and the supervised fine-tuned policy $\pi_{ref}$*

**Theoretical Insights and Key Remarks :** The sub-optimality gap is expressed in terms of the difference between the target reward function and the reward function of the best model in our policy set (closest to $r_{\text{target}}$) $\delta_{*j}$ for each token $\mathbf{s}_t$ and the sum of KL divergences w.r.t the reference policy at the token and trajectory level. The sub-optimality gap thus guarantees that the performance of our multi-agent decoding strategy will improve over the best-performing policy (closest to the

Table 1: **GPT-4 Based Evaluation.** We prompt GPT-4 to rate responses from various decoding strategies on relevance, accuracy, and insightfulness, scoring them from 1 to 10. A higher win-tie percentage indicates our method's effectiveness in generating contextually relevant and accurate responses.

| | | | Win-Tie (%) ↑ | | | | | | |
|---|---|---|---|---|---|---|---|---|---|
| **Ours** | vs. | **Methods** | | Task-I | | | | Task-II | |
| | | | **Evaluation-1** | **Evaluation-2** | **Evaluation-3** | **Evaluation-4** | **Evaluation-5** | **Evaluation-6** | **Evaluation-7** |
| COLLAB | Agent-I | | 70.78 | 71.89 | 69.38 | 60.00 | 57.14 | 64.28 | 59.42 |
| COLLAB | Agent-II | | 67.41 | 63.56 | 55.10 | 65.00 | 58.69 | 61.05 | 71.92 |
| COLLAB | BoN Sampling | | 68.54 | 69.24 | 61.22 | 66.32 | 50.00 | 73.75 | 65.00 |

target LLM policy) from our policy set, under the assumptions. The gap decreases as $\delta_{*j}$ becomes smaller, meaning that the reward function of the best LLM agent in the set closely aligns with the target reward. Furthermore, by adjusting the regularization constants $\alpha, \beta$ we can control the trade-off between closeness to the reference policy and distance from the optimal policy. The term $\beta KL(\rho^{\pi^*}(\cdot|\mathbf{s}_t), \rho_{\text{ref}}(\cdot|\mathbf{s}_t))$ is constant and independent of $j$ quantifies the divergence between the optimal policy and the reference policy, which is lower in two cases 1) $\beta$ is small and 2) when the reference policy is closer to the optimal policy. Thus, the sub-optimality gap will be lower when the best agent's reward function is close to the target reward function, and when the regularization terms are properly controlled to maintain proximity to both the reference policy and the optimal policy. In contrast to prior results in decoding and alignment (Rafailov et al., 2023; Mudgal et al., 2024; Chakraborty et al., 2024b), we mainly incur an additional term $\delta_{*j}$ in the upper bound. However, as discussed prior alignment approaches either require fine-tuning billion parameters ((Rafailov et al., 2023)) for each new $r_{\text{target}}$ which is expensive, or access to an aligned policy under $r_{\text{target}}$ (decoding (Mudgal et al., 2024)) which might be unavailable. Thus in the absence of the aligned policy to the target reward, there will be a persistent gap $|r_j - r_{\text{target}}|$ for single agent decoding, which is improved using the diverse mixture of agents.

## 5 EXPERIMENTAL EVALUATIONS

In this section, we present a comprehensive empirical analysis of our proposed framework, tested across various open-source datasets and state-of-the-art models (Lambert et al., 2024). Our findings demonstrate COLLAB's effectiveness in aligning language model outputs with specific target rewards. For implementation, we set the number of tokens sampled (top-p) $p = 10$ and the decoding alignment parameter $\alpha = 1$. Reproducibility is ensured through the use of publicly available resources.

**Experiment Setup:** To demonstrate the efficacy of our proposed multi-agent decoding framework COLLAB, we utilize several open-source LLMs fine-tuned on distinct and diverse capabilities, as detailed in Table 3. Our multi-agent decoding framework is evaluated across 7 distinct setups, as illustrated in Table 3:

1. Evaluation-1 to Evaluation-4 (Task-I): For this task, we utilize the Berkeley Nectar dataset (Zhu et al., 2023) to test the agent's capacity for multi-turn dialogues and question answering.

2. Evaluation-5 to Evaluation-7 (Task-II): We employ the HH-RLHF dataset (Bai et al., 2022) to assess the agent's helpfulness and ethical alignment in response generation.

**Evaluation Methodology & Metrics.** For evaluation, we compare the performance of the response generated by the language model corresponding to each prompt in the test dataset. Following (Khanov et al., 2024; Chakraborty et al., 2024b), we limit the maximum length of the prompt and generated continuation to 128 and 2048 tokens, respectively. For all baselines, we utilize a greedy-based sampling method. The quality of the generated responses is assessed based on:

• **Average Reward:** We report the mean of the rewards for generations corresponding to all prompts in the test set. A higher mean reward score signifies that the model's outputs are better aligned with the attributes represented in the target reward model.

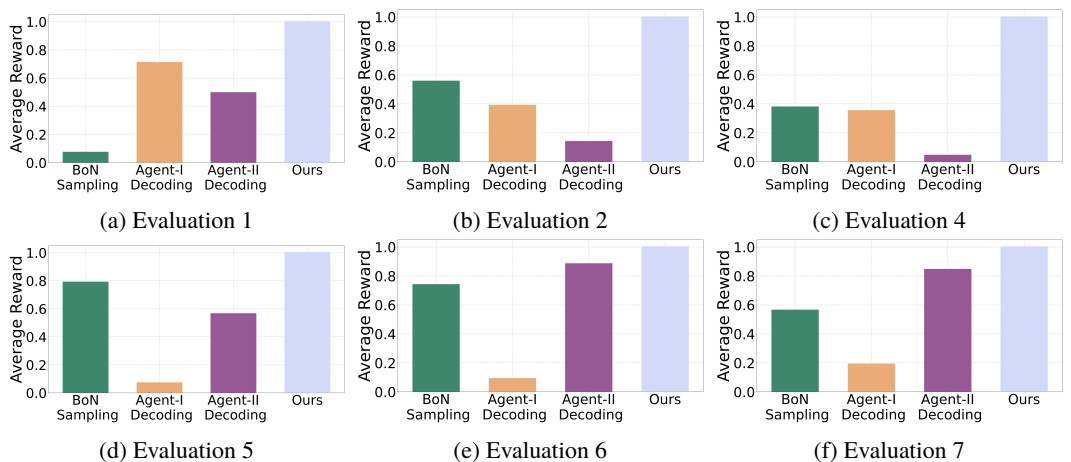

Figure 2: In the above plots, we present the normalized average reward values obtained using the corresponding setup outlined in Table 3. Agent-I, and Agent-II refers to the average reward obtained by the individual models with SoTA decoding. For the BoN agents sampling, we perform vanilla logit-based sampling using individual agents and select the best response w.r.t the target reward. Our analysis reveals that across all setups, CoLLAB consistently outperforms other baselines summarized in Table 3, demonstrating the importance of multi-agent decoding.

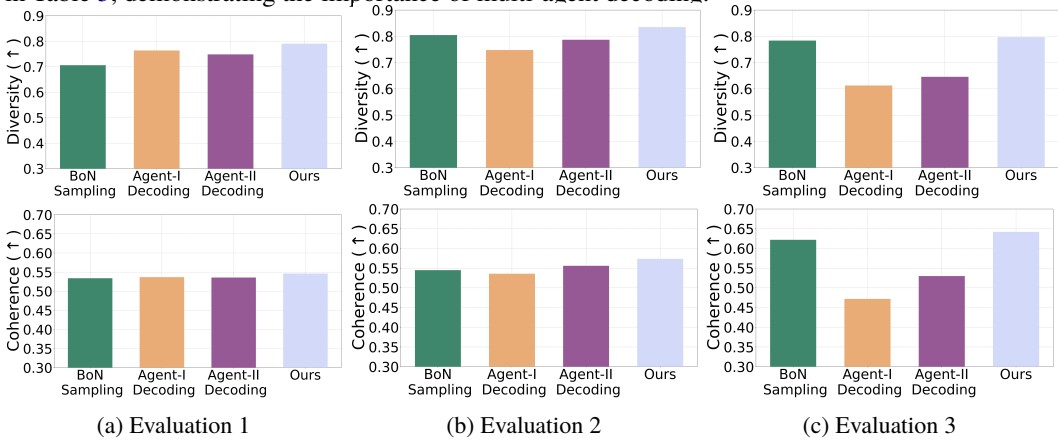

Figure 3: In the above plots, we present the diversity and coherence values obtained using the corresponding setup as outlined in Table 3. We clearly observe the response generated using CoLLAB consistently outperforms other baselines in-terms of both diversity and coherence. This also indicates switching between agents with Implicit-Q helps in improving the overall quality of the responses

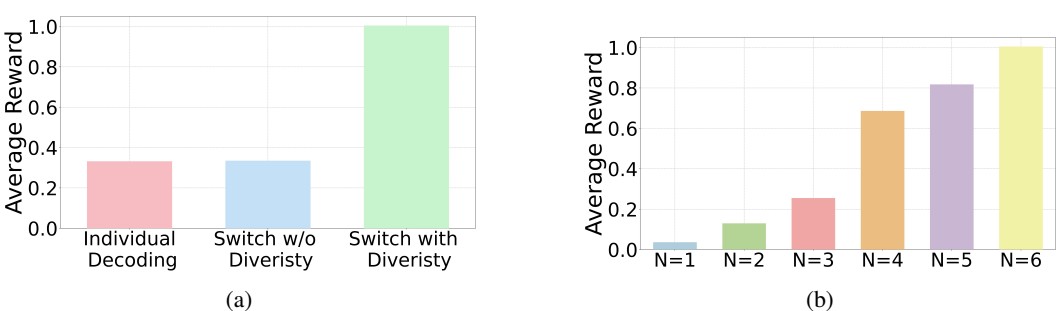

Figure 4: **Left.** The bar plot highlights the importance of using diverse agents for enhanced decoding; employing different but non-diverse agents results in poor performance. **Right.** The visualization shows the improvement in average reward as the number of diverse agents increases.

- **GPT-4 Winrate:** To further evaluate the quality of the generated responses, we employ a GPT-4-based evaluation framework, using GPT-4 as a surrogate for human assessment. We prompt GPT-4 to rate responses from various decoding strategies on relevance, accuracy, and insightfulness, scoring them from 1 to 10. A higher win-tie percentage indicates our method's effectiveness in generating contextually relevant and accurate responses. This is also considered as a surrogate for the target reward as common in (Rafailov et al., 2023; OpenAI et al., 2024).

- **Diversity:** This metric measures the ability to generate texts with a wide range of vocabulary. Specifically, we calculate diversity using the frequency of repeated n-grams in the generated text. A lower repetition rate signifies higher diversity, indicating that the model is capable of generating novel, varied responses instead of repeating phrases or structures frequently.

- **Coherence:** Coherence evaluates the semantic relatedness between each prompt and its generated response. We use SimCSE-generated (Su et al., 2022) embeddings to represent both the prompt and the response. The cosine similarity between these embeddings is then calculated to quantify their semantic closeness. A higher cosine similarity score indicates that the generated response is contextually relevant and coherent with the input prompt.

**Baselines.** In Figure 2, we present the normalized average rewards for six setups detailed in Table 3. We compare our proposed method COLLAB with comprehensive baseline approaches including decoding with individual agents and BoN sampling (Nakano et al., 2021). For generating baseline responses, we leverage state-of-art decoding approaches (Khanov et al., 2024; Chakraborty et al., 2024b). Specifically, in our experimental setting, Agent-1, Agent-2 represent the response generated using SoTA decoding (Chakraborty et al., 2024b). For the BoN agents sample, we perform vanilla logit-based sampling using individual agents and select the best response w.r.t the target reward. To provide a clearer comparison of results, we normalize the average rewards (further details of normalization in Appendix E.1).

**Evaluation Results.** In our experimental evaluations, we demonstrate that our proposed framework, COLLAB, consistently outperforms existing baselines across all setups and a diverse range of metrics. As shown in Figure 2, our approach achieves superior performance in terms of average reward alignment with the target model. Notably, we observe that simply switching between two LLM agents allows the resulting decoding strategy to generate responses that achieve significantly higher rewards than either individual agent, reinforcing our point. To further evaluate the quality of the generated responses, we report the diversity and coherence metrics in Figure 3 for three evaluation setups. The responses generated using COLLAB outperform all other baselines across all evaluation metrics.

**GPT-4 Evaluation and Insights.** To assess the performance of our approach, we employed GPT-4 to evaluate and rate pairs of responses to the same prompt on a scale from 1 to 10, focusing on relevance, accuracy, helpfulness, harmlessness, and insightfulness. We randomly sampled 300 prompts from the test set and compared the responses generated by COLLAB with those from other competitive decoding methods. The GPT-4 evaluation results are presented in Table 1, showing the percentage of win-ties for our method over baseline decoding strategies. A higher percentage indicates that our proposed method is more proficient in generating responses that align better with human preferences. As shown in Table 1, COLLAB consistently achieves a higher win-tie percentage compared to other decoding approaches, reaffirming its efficacy.

**Effect of Increasing the Number and Diversity of Agents in the Mixture.** Figure 4a presents visualizations illustrating the impact of agent diversity on decoding performance. For this evaluation, we consider two setups: (1) Switch without Diversity, where we use two similar models for switching; and (2) Switch with Diversity, where we employ two diverse agents. The results indicate that using a diverse mixture of agents significantly improves the average reward compared to individual decoding and switching between non-diverse agents. Furthermore, Figure 4b demonstrates that as the number of diverse agents increases, there is a consistent improvement in average reward, highlighting the importance of both the number and diversity of agents for enhancing decoding performance.

ACKNOWLEDGMENTS

Huang is supported by DARPA Transfer from Imprecise and Abstract Models to Autonomous Technologies (TIAMAT) 80321, National Science Foundation NSF-IIS-2147276 FAI, DOD-ONR-Office

of Naval Research under award number N00014-22-1-2335, DOD-AFOSR-Air Force Office of Scientific Research under award number FA9550-23-1-0048, DOD-DARPA-Defense Advanced Research Projects Agency Guaranteeing AI Robustness against Deception (GARD) HR00112020007, Adobe, Capital One and JP Morgan faculty fellowships. The authors would like to thank Amrit Singh Bedi for helpful discussions during the problem formulation.

## DISCLAIMER

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

## A  SOFTWARE AND HARDWARE

We run all experiments with Python 3.7.4 and PyTorch 1.9.0. For all experimentation, we use two Nvidia RTX A6000 GPUs.

## B  NOTATIONS

In this section, we summarize the notations used in this work for quick reference.

Table 2: **Notations.** This table presents the notations we used for this work.

| Implication | Notation |
|---|---|
| Vocabulary | $\mathcal{V}$ |
| Prompt | $\mathbf{x}$ |
| Response | $\mathbf{y}$ |
| Token $t$ in response | $y_t$ |
| End of Sentence token | EOS |
| $a$ concatenation $b$ | $[a, b]$ |
| Response till token $t$ | $\mathbf{y}_{<t} := [y_1, y_2, \cdots, y_{t-1}]$ |
| Reward function | $r(\mathbf{x}, \mathbf{y})$ |
| Token level state | $\mathbf{s}_t := [\mathbf{x}, \mathbf{y}_{<t}]$ |
| Token level action | $a_t$ |
| Trajectory level LLM policy | $\rho(\cdot\|\mathbf{x})$ |
| Token level LLM policy | $\pi(\cdot\|\mathbf{s}_t)$ |
| Action value function corresponding to $R$ under policy $\pi$ | $Q^\pi$ |
| Value function corresponding to $R$ under policy $\pi$ | $V^\pi$ |

## C  RELATED WORKS

With the increasing complexity and wide-spread use of the generative models, it is crucial to ensure that such models are well aligned with human preferences, social welfare and broader goals especially in critical decision making scenarios. Alignment via fine-tuning with reinforcement learning from human feedback (RLHF) has emerged as a key paradigm for aligning foundation models (Ouyang et al., 2022; Chakraborty et al., 2024a; Rafailov et al., 2023; Chakraborty et al., 2024c; Chen et al., 2024). RLHF typically operates in two phases: first, a reward model is trained on human feedback, and then a policy is fine-tuned with reinforcement learning (PPO (Schulman et al., 2017)) using the trained reward model (Ouyang et al., 2022; Stiennon et al., 2022; Ziegler et al., 2020; Yuan et al., 2023; Go et al., 2023; Vamplew et al., 2018; 2008). Direct preference optimization seeks to stabilize the training of preference alignment through reductions to supervised learning training (Rafailov et al., 2023). Although these training methods have proven effective in aligning generative models, they remain computationally demanding and assume white-box access to the model parameters which is not true in many industry applications. Specifically, fine-tuning billions of parameters for a new task is either prohibitively costly or requires overriding enterprise constraints on model availability. This makes RL or supervised learning (DPO)-based fine-tuning for alignment unsuitable for obtaining reasonable performance on new preferences or tasks.

On the other hand, decoding-based methods (Mudgal et al., 2024; Chakraborty et al., 2024b) have emerged as an alternate way of alignment without fine-tuning the model parameters. Decoding operates by altering the distribution of the generated response to align to the target preference directly without updating the parameters of the LLM. The work by (Mudgal et al., 2024) is one of the first to integrate the alignment procedure directly into the decoding process, where they propose adjusting the generation probabilities at each decoding step based on feedback from a reward model. (Huang et al., 2024) redefined the text-generation process as a search problem, with LLMs acting as search agents and they employ a heuristic-guided search mechanism to generate responses based on a given prompt. The most recent research around Controlled and Principled decoding (CD, TQ*) formulates the decoding problem as a KL regularized Reinforcement learning problem and obtaining a closed-form solution with an estimate of $Q^*$ for decoding.

However, these prior methods all are predicated upon a single reward model (or task specification), and thus cannot incorporate a diversity of constituent models and their associated knowledge bases. Hence, they mostly lack a concrete way to handle tasks significantly different from their training data. This motivates us to design an efficient strategy to optimally synthesize information contained in a diversity of constituent LLMs to align to a target preference via decoding.

## D   THEORETICAL RESULTS AND INSIGHTS

In this section, we characterize the sub-optimality of our proposed approach below. First, our algorithm's policy is defined by

$$\pi_{\text{alg}} \in \arg\max_z \max_j J_{\text{target}}^{\pi_j}(\mathbf{s_t}, z) \tag{10}$$

where, $J_{\text{target}}^{\pi_j}(\mathbf{s_t}, z) = Q_{\text{target}}^{\pi_j}(\mathbf{s_t}, z) - \alpha KL(\pi_j(\cdot|\mathbf{s_t}), \pi_{\text{ref}}(\cdot|\mathbf{s_t}))$

From the definition above, we know that

$$J_{\text{target}}^{\pi_{\text{alg}}}(\mathbf{s_t}, z) \geq \max_j J_{\text{target}}^{\pi_j}(\mathbf{s_t}, z) \tag{11}$$

**Lemma 1.** *For any two policies $\pi_i$ and $\pi_j$, the difference between their Q-values at a state-action pair $(\mathbf{s_t}, z)$ can be bounded as:*

$$Q_i^{\pi_i}(\mathbf{s_t}, z) - Q_j^{\pi_j}(\mathbf{s_t}, z) \leq \delta_{ij} + \beta KL(\rho^{\pi_i}(\cdot|\mathbf{s_t}, z), \rho_{ref}(\cdot|\mathbf{s_t}, z)) - \beta KL(\rho^{\pi_j}(\cdot|\mathbf{s_t}, z), \rho_{ref}(\cdot|\mathbf{s_t}, z))$$

*where $\delta_{ij} = \max_\tau |r_j([\mathbf{s_t}, z], \tau) - r_j([\mathbf{s_t}, z], \tau)|$ and $\beta$ is a regularization coefficient.*

**Proof** : We start by considering the difference between the Q-values for policies $\pi_i$ and $\pi_j$ at state action pair $(\mathbf{s_t}, z)$:

$$Q_i^{\pi_i}(\mathbf{s_t}, z) - Q_j^{\pi_j}(\mathbf{s_t}, z) = \underbrace{Q_i^{\pi_i}(\mathbf{s_t}, z) - Q_j^{\pi_i}(\mathbf{s_t}, z)}_{T_1} + \underbrace{Q_j^{\pi_i}(\mathbf{s_t}, z) - Q_j^{\pi_j}(\mathbf{s_t}, z)}_{T_2} \tag{12}$$

where we first add and subtract the term $Q_j^{\pi_i}(\mathbf{s_t}, z)$ and separately analyze the two terms $T_1$ and $T_2$.

$$\begin{aligned} T_1 &= Q_i^{\pi_i}(\mathbf{s_t}, z) - Q_j^{\pi_i}(\mathbf{s_t}, z) \\ &= \mathbb{E}_{\tau \sim \rho^{\pi_i}(\cdot|\mathbf{s_t}, z)}[r_i([\mathbf{s_t}, z], \tau)] - \mathbb{E}_{\tau \sim \rho^{\pi_i}(\cdot|\mathbf{s_t}, z)}[r_j([\mathbf{s_t}, z], \tau)] \\ &\leq \max_\tau |r_i([\mathbf{s_t}, z], \tau) - r_j([\mathbf{s_t}, z], \tau)| \end{aligned} \tag{13}$$

where the first term $T_1$ can be upper-bounded by $T_1 \leq \max_\tau |r_i([\mathbf{s_t}, z], \tau) - r_j([\mathbf{s_t}, z], \tau)|$ which we denote as $\delta_{ij}$ for simplicity. Next we proceed to bound the term $T_2$ as:

$$\begin{aligned} T_2 &= Q_j^{\pi_i}(\mathbf{s_t}, z) - Q_j^{\pi_j}(\mathbf{s_t}, z) \\ &= \mathbb{E}_{\tau \sim \rho^{\pi_i}(\cdot|\mathbf{s_t}, z)}[r_j([\mathbf{s_t}, z], \tau)] - \mathbb{E}_{\tau \sim \rho^{\pi_j}(\cdot|\mathbf{s_t}, z)}[r_j([\mathbf{s_t}, z], \tau)] \\ &\quad + \beta KL(\rho^{\pi_j}(\cdot|\mathbf{s_t}, z), \rho_{\text{ref}}(\cdot|\mathbf{s_t}, z)) - \beta KL(\rho^{\pi_j}(\cdot|\mathbf{s_t}, z), \rho_{\text{ref}}(\cdot|\mathbf{s_t}, z)) \\ &\quad + \beta KL(\rho^{\pi_i}(\cdot|\mathbf{s_t}, z), \rho_{\text{ref}}(\cdot|\mathbf{s_t}, z)) - \beta KL(\rho^{\pi_i}(\cdot|\mathbf{s_t}, z), \rho_{\text{ref}}(\cdot|\mathbf{s_t})) \\ &\leq \beta KL(\rho^{\pi_i}(\cdot|\mathbf{s_t}, z), \rho_{\text{ref}}(\cdot|\mathbf{s_t}, z)) - \beta KL(\rho^{\pi_j}(\cdot|\mathbf{s_t}, z), \rho_{\text{ref}}(\cdot|\mathbf{s_t}, z)) \end{aligned} \tag{14}$$

where, we first expand upon the definition of Q-function and add and subtract the KL-divergence of the individual policies $\rho^{\pi_i}, \rho^{\pi_j}$. Next, we leverage the optimality of the KL regularized policy for the alignment objective. Since, we know $\rho_j$ is the optimal policy for the beta KL regularized objective (Ouyang et al., 2022; Rafailov et al., 2023) under the reward function $r_j([\mathbf{s_t}, z], \tau)$ i.e $\mathbb{E}_{\tau \sim \rho^{\pi_j}(\cdot|\mathbf{s_t}, z)}[r_j([\mathbf{s_t}, z], \tau)] - \beta KL(\rho^{\pi_j}(\cdot|\mathbf{s_t}, z), \rho_{\text{ref}}(\cdot|\mathbf{s_t}, z))$ is greater than $\mathbb{E}_{\tau \sim \rho^{\pi_i}(\cdot|\mathbf{s_t}, z)}[r_j([\mathbf{s_t}, z], \tau)] - \beta KL(\rho^{\pi_i}(\cdot|\mathbf{s_t}, z), \rho_{\text{ref}}(\cdot|\mathbf{s_t}, z))$ for all $i$. Thus leveraging this optimality condition, we obtain the desired result:

$$Q_i^{\pi_i}(\mathbf{s_t}, z) - Q_j^{\pi_j}(\mathbf{s_t}, z) \leq \delta_{ij} + \beta KL(\rho^{\pi_i}(\cdot|\mathbf{s_t}, z), \rho_{\text{ref}}(\cdot|\mathbf{s_t}, z)) - \beta KL(\rho^{\pi_j}(\cdot|\mathbf{s_t}, z), \rho_{\text{ref}}(\cdot|\mathbf{s_t}, z))$$

This completes the proof.

## D.1 PROOF OF THEOREM 1

As discussed, $\Pi = \{\pi_1, \pi_2, \ldots, \pi_K\}$ is the set of pre-trained policies, each aligned to a latent reward function $r_j$, and $\pi_{\text{alg}}$ be the policy obtained by the mixture of agents decoding strategy. Assume that the optimal policy for the target reward function $r_{\text{target}}$ is $\pi^*$. Then, the sub-optimality of the mixture of agents decoding policy $\pi_{\text{alg}}$ with respect to the optimal policy $\pi^*$ is bounded by:

$$\Delta \leq \min_{j \in K} \delta_{*j} + \alpha \left[ KL(\pi_j(\cdot|s), \pi_{\text{ref}}(\cdot|s)) \right] + \beta KL(\rho^{\pi^*}(\cdot|s), \rho_{\text{ref}}(\cdot|s)) \tag{15}$$

where $\delta_{*j} = \max_\tau |r_{\text{target}}([\mathbf{s_t}, z], \tau) - r_j([\mathbf{s_t}, z], \tau)|$

**Proof**: We begin by defining the sub-optimality gap in Q-function as

$$\Delta = Q_{\text{target}}^{\pi^*}(\mathbf{s_t}, z) - Q_{\text{target}}^{\pi_{\text{alg}}}(\mathbf{s_t}, z) \tag{16}$$

where this sub-optimality gap denotes the performance gap of our algorithm against the optimal policy for the target reward $r_{\text{target}}$.

$$\begin{aligned}
\Delta &= Q_{\text{target}}^{\pi^*}(\mathbf{s_t}, z) - Q_{\text{target}}^{\pi_{\text{alg}}}(\mathbf{s_t}, z) \\
&= Q_{\text{target}}^{\pi^*}(\mathbf{s_t}, z) - Q_j^{\pi_j}(\mathbf{s_t}, z) + Q_j^{\pi_j}(\mathbf{s_t}, z) - Q_{\text{target}}^{\pi_{\text{alg}}}(\mathbf{s_t}, z) \\
&= \Delta_1 + \Delta_2
\end{aligned} \tag{17}$$

where we add and subtract $Q_j^{\pi_j}(\mathbf{s_t}, z)$ and then separate the two components as $\Delta_1, \Delta_2$. We first derive an upper-bound on $\Delta_1$ as

$$\begin{aligned}
\Delta_1 &= Q_{\text{target}}^{\pi^*}(\mathbf{s_t}, z) - Q_j^{\pi_j}(\mathbf{s_t}, z) \\
&\leq \delta_{*j} + \beta KL(\rho^{\pi^*}(\cdot|\mathbf{s_t}, z), \rho_{\text{ref}}(\cdot|\mathbf{s_t}, z)) - \beta KL(\rho^{\pi_j}(\cdot|\mathbf{s_t}, z), \rho_{\text{ref}}(\cdot|\mathbf{s_t}, z))
\end{aligned} \tag{18}$$

where, $\delta_{*j} = \max_\tau |r_{\text{target}}([\mathbf{s_t}, z], \tau) - r_j([\mathbf{s_t}, z], \tau)|$. We obtained the bound by directly leveraging the result from Lemma 1 for the reward functions $r_{\text{target}}(\mathbf{s_t}, z), r_j(\mathbf{s_t}, z)$. Next, we proceed to bound the term $\Delta_2$ as

$$\begin{aligned}
\Delta_2 &= Q_j^{\pi_j}(\mathbf{s_t}, z) - Q_{\text{target}}^{\pi_{\text{alg}}}(\mathbf{s_t}, z) \\
&= Q_j^{\pi_j}(\mathbf{s_t}, z) - Q_{\text{target}}^{\pi_{\text{alg}}}(\mathbf{s_t}, z) + \alpha KL(\pi_{\text{alg}}(\cdot|\mathbf{s_t}), \pi_{\text{ref}}(\cdot|\mathbf{s_t})) - \alpha KL(\pi_{\text{alg}}(\cdot|\mathbf{s_t}), \pi_{\text{ref}}(\cdot|\mathbf{s_t})) \\
&= Q_j^{\pi_j}(\mathbf{s_t}, z) - J_{\text{target}}^{\pi_{\text{alg}}}(\mathbf{s_t}, z) - \alpha KL(\pi_{\text{alg}}(\cdot|\mathbf{s_t}), \pi_{\text{ref}}(\cdot|\mathbf{s_t})) \\
&\leq Q_j^{\pi_j}(\mathbf{s_t}, z) - Q_{\text{target}}^{\pi_j}(\mathbf{s_t}, z) + \alpha KL(\pi_j(\cdot|\mathbf{s_t}), \pi_{\text{ref}}(\cdot|\mathbf{s_t})) - \alpha KL(\pi_{\text{alg}}(\cdot|\mathbf{s_t}), \pi_{\text{ref}}(\cdot|\mathbf{s_t})) \\
&\leq \min_{j \in K} \delta_{*j} + \alpha KL(\pi_j(\cdot|s), \pi_{\text{ref}}(\cdot|s)) - \alpha KL(\pi_{\text{alg}}(\cdot|s), \pi_{\text{ref}}(\cdot|s))
\end{aligned} \tag{19}$$

where, first we add and subtract the Kl divergence terms $\alpha KL(\pi_{\text{alg}}(\cdot|\mathbf{s_t})$ and next utilizing the fact that $J_{\text{target}}^{\pi_{\text{alg}}}(\mathbf{s_t}, z) = Q_{\text{target}}^{\pi_{\text{alg}}}(\mathbf{s_t}, z) - \alpha KL(\pi_{\text{alg}}(\cdot|\mathbf{s_t}), \pi_{\text{ref}}(\cdot|\mathbf{s_t}))$, we get the third equation. Finally, we know that for any $j \in K$, $J_{\text{target}}^{\pi_{\text{alg}}}(\mathbf{s_t}, z) \geq J_{\text{target}}^{\pi_j}(\mathbf{s_t}, z)$ and utilizing this we get the final inequality. Since it holds for any $j$, the final inequality holds true, where $\delta_{*j} = $ Now, combining the upper-bounds from $\Delta_1, \Delta_2$, we get

$$\Delta \leq \min_{j \in K} \delta_{*j} + \alpha KL(\pi_j(\cdot|\mathbf{s_t}), \pi_{\text{ref}}(\cdot|\mathbf{s_t})) + \beta KL(\rho^{\pi^*}(\cdot|\mathbf{s_t}), \rho_{\text{ref}}(\cdot|\mathbf{s_t})) \tag{20}$$

where the above bounds express the sub-optimality of our proposed approach in terms of the difference between the target reward function and the reward function of the best model in our policy set for each token $\mathbf{s_t}$ and the sum of KL divergences w.r.t the reference policy at the token and trajectory level. This completes the proof.

## E ADDITIONAL EXPERIMENTAL DETAILS

### E.1 REWARD NORMALIZATION

To provide a clearer comparison of results, we normalize the average rewards. For example: let $r_{\text{Agent-I}}$ represent the average reward achieved by the Agent-I model across all generated responses to the test prompts. The normalized reward, $\tilde{r}_{\text{Agent-I}}$, is calculated as $\tilde{r}_{\text{Agent-I}} = \frac{r_{\text{Agent-I}} - r_{\text{MIN}}}{r_{\text{COLLAB}} - r_{\text{MIN}}}$, where $r_{\text{MIN}}$ is the minimum reward obtained by the model for a given specific dataset. This ensures that the results are scaled relative to existing methods.

### E.2 Setups

Table 3: Summary of the datasets and model architectures used for experimental evaluations.

| Target Task | Setup | Dataset | Model Architectures | | |
|---|---|---|---|---|---|
| | | | **Agent-I** | **Agent-II** | **Reward** |
| Task-I | Evaluation-1 | Berkeley Nectar (Zhu et al., 2023) | Zephyr-7B-$\alpha$ | Starling-7B-$\alpha$ | Mistral-7B-$\alpha$-IT |
| | Evaluation-2 | Berkeley Nectar (Zhu et al., 2023) | Zephyr-7B-$\alpha$ | Zephyr-7B-$\beta$ | Mistral-7B-$\alpha$-IT |
| | Evaluation-3 | Berkeley Nectar (Zhu et al., 2023) | Zephyr-Qwen-2-7B-DPO | Dolphin-Qwen-2-7B | Mistral-7B-$\alpha$-IT |
| | Evaluation-4 | Berkeley Nectar (Zhu et al., 2023) | Dolphin-2.6-Mistral-7B-DPO | Starling-7B-$\alpha$ | Mistral-7B-$\alpha$-IT |
| Task-II | Evaluation-5 | HH-RLHF (Bai et al., 2022) | Zephyr-7B-$\alpha$ | Starling-7B-$\alpha$ | Mistral-7B-$\alpha$ |
| | Evaluation-6 | HH-RLHF (Bai et al., 2022) | Dolphin-2.6-Mistral-7B-DPO | Starling-7B-$\alpha$ | Mistral-7B-$\alpha$ |
| | Evaluation-7 | HH-RLHF (Bai et al., 2022) | Zephyr-Qwen-2-7B-DPO | Dolphin-Qwen-2-7B | Mistral-7B-$\alpha$ |

## F  Motivation for Greedy Switching Policy

Let $s_t = [x, y_{\leq t-1}]$ denote the state made up of the prompt and the sequentially generated tokens. Recall the $L-$length average target reward for a policy $\pi$ given as

$$Q^{\pi}_{\text{target}}(s, a) = \mathbb{E}^{\pi}\left[\sum_{t=0}^{L-1} r_{\text{target}}(s_t, a_t)\Big| s_t = s, a_t = a \sim \pi(\cdot|s_t)\right]. \tag{21}$$

In our problem, we do not have an optimal policy for $r_{\text{target}}$, and we have to do the best we can using a collection of policies $\Pi = \{\pi_1, \cdots, \pi_K\}$.

- Consider decoding just the last token. For a give state $s_T$, we need to find a single token (action) that maximizes the target reward. A best policy selection is given as

$$\pi_T^{\text{best}}(\cdot|s_T) = \arg\max_{\pi_j \in \Pi} \mathbb{E}^{\pi_j}\left[r_{\text{target}}(s_T, a \sim \pi_j(\cdot|s_T))\right].$$

- Consider decoding the last two tokens. For a given state $s_{T-1}$, a best policy selection is given as

$$\pi_{T-1}^{\text{best}}(\cdot|s_{T-1}) = \arg\max_{\pi_j \in \Pi} \mathbb{E}^{\pi_j}\left[r_{\text{target}}(s_{T-1}, a \sim \pi_j(\cdot|s_{T-1})) + \mathbb{E}_s\left\{\max_{\pi_j \in \Pi} \mathbb{E}^{\pi_j}\left[r_{\text{target}}(s, a \sim \pi_j(\cdot|s))\right]\Big|s_{T-1}\right\}\right],$$

where $\mathbb{E}_s\{\cdot\}$ is the expectation over the next states.

Based on this define the $\mathcal{Q}-$function iteratively as follows:

- $\mathcal{Q}_T(s_T, a_T) = \max_{\pi_j \in \Pi} \mathbb{E}^{\pi_j}\left[r_{\text{target}}(s_T, a \sim \pi_j(\cdot|s_T))\right]$

- For $l = T-1, \cdots, 0$, we have

$$\mathcal{Q}_l(s_l, a_l) = \max_{\pi_j \in \Pi} \mathbb{E}^{\pi_j}\left[r_{\text{target}}(s_l, a_l \sim \pi_j(\cdot|s_l))\right. \tag{22}$$

$$\left. + \mathbb{E}_{s_{l+1}} \max_{\pi_j \in \Pi} \mathcal{Q}_{l+1}(s_{l+1}, a_{l+1} \sim \pi_j(\cdot|s_{l+1}))\right]. \tag{23}$$

As seen from equation 22, finding the best policy at every step based on dynamic programming becomes infeasible for LLMs. Therefore, as an easily decodable greedy policy, we use the following crude proxy:

$$\mathbb{E}^{\pi_j}\left[r_{\text{target}}(s_l, a_l \sim \pi_j(\cdot|s_l)) + \mathbb{E}_{s_{l+1}} \max_{\pi \in \Pi} \mathcal{Q}_{l+1}(s_{l+1}, a_{l+1} \sim \pi(\cdot|s_{l+1}))\right] \approx Q^{\pi_j}_{\text{target}}(s_l, a_l),$$

whence the approximate best policy selection now becomes,

$$\pi_k^{\text{approx. best}}(\cdot|s_k) = \arg\max_{\pi_j \in \Pi} Q^{\pi_j}_{\text{target}}(s_k, a_k \sim \pi_j(\cdot|s_k)).$$

## G  Examples of the Generated Text of our Algorithm compared to Baselines

In this section, we show more examples of text generated using different decoding strategies:

---

**Example-1**

**Prompt:** 0 looks like O and 1 looks like I. What does 2 look like?

---

**Agent-I:** The number 2 can be seen as the letter B, as the two horizontal lines are reminiscent of the upper and lower parts of the letter B, while the vertical line in the middle is reminiscent of the vertical line in the middle of the letter B. In addition, some people might also see the number 2 as the letter E, as the two horizontal lines can be seen as the top and bottom of the letter E, while the vertical line in the middle can be seen as the vertical line in the middle of the letter E. However, this interpretation is less common.

---

**Agent-II:** Out of the given options, the one that fits this sentence is older (option 2). Thus, the correct option is (2)

---

**BoN Sampling:** 2 looks like the numeral "2". In the context of binary code, where each digit represents a binary value (0 or 1), the number 2 would be represented as "10" in binary. However, when discussing numerals or digits in general, 2 is represented as the symbol "2". In some contexts, the word "two" might be used to represent the number 2, but this is less common when discussing numerals or digits.

---

**Collab (Ours):** The specific representation of 2 can vary depending on the context or writing system. In the decimal numeral system, it is commonly represented as a single character 2, which is a horizontal line with two vertical strokes. In other systems, such as the ancient Egyptian hieroglyphic system, 2 might be represented as the hieroglyph for "two" which is a curved line with a vertical line or a single character that looks like a "T" rotated 90 degrees.

---

---

**Example-2**

**Prompt:** NAME-1 in Jordan is [blank] than Machu Picchu in Peru. (1) old (2) older(3) elder (4) eldest

---

**Agent-I:** The number 2 can be seen as the letter B, as the two horizontal lines are reminiscent of the upper and lower parts of the letter B, while the vertical line in the middle is reminiscent of the vertical line in the middle of the letter B. In addition, some people might also see the number 2 as the letter E, as the two horizontal lines can be seen as the top and bottom of the letter E, while the vertical line in the middle can be seen as the vertical line in the middle of the letter E. However, this interpretation is less common.

---

**Agent-II:** NAME-1 in Jordan is older than Machu Picchu in Peru. Explanation: The correct answer is (2) older. When comparing two things, we use older to describe the one that is more aged or has existed for a longer period of time. In this case, NAME-1 in Jordan is being compared to Machu Picchu in Peru, and since NAME-1 is older, the correct term to use is older.

---

**BoN Sampling:** The correct answer is older because it is the most appropriate term to describe the age difference between the two sites. Nabataean city of Petra, also known as NAME-1, in Jordan is older than Machu Picchu in Peru. Petra, the ancient city of the Nabataeans, is located in present-day Jordan and is known for its rock-cut architecture and water conduit system. It was established in the 4th century BCE and reached its peak during the 1st century CE.

---

**COLLAB (Ours):** older, Jerash, also known as the Pompeii of the East or Gerasa is an ancient city in Jordan. Although Jerash is older than Machu Picchu, the most accurate term to compare their ages is older. Machu Picchu, a 15th-century Inca citadel, was built in the mid-to-late 1400s. Jerash, on the other hand, is believed to have been founded around the 3rd century BC and has a rich history that spans various civilizations.

---

**Example-3**

**Prompt:** 1.5 chickens lay 1.5 eggs in 1.5 days. How many eggs will 9 chickens lay in 9 days?

---

**Agent-I:** To find the total number of eggs laid by 9 chickens in 9 days, we need to multiply the number of eggs laid per chicken per day (1.5) by the number of chickens and the number of days: Eggs = (1.5 eggs per chicken per day) x (9 chickens) x (9 days) Eggs = 135 eggs So, 9 chickens will lay 135 eggs in 9 days. The answer: 135 eggs.

---

**Agent-II:** We need to determine how many eggs one chicken lays in one day. Since 1.5 chickens lay 1.5 eggs in 1.5 days, this means that in one day, one chicken will lay one egg. Therefore, if we have nine chickens, they will lay a total of 9 * 1 = 9 eggs in one day. Since we want to know how many eggs the chickens will lay in nine days, we simply multiply the number of eggs laid in one day (9) by the number of days (9). This gives us a total of 9 * 9 = 81 eggs.

---

**COLLAB (Ours):** We need to find the total number of eggs laid by 9 chickens in 9 days, we need to start by determining how many eggs one chicken lays in one day. From the given, 1.5 chickens lay 1.5 eggs in 1.5 days. This means that 1 chicken lays 1 egg in 1.5 days. To find the number of eggs one chicken lays in one day: Eggs per day = 1 egg / 1.5 days = 2/3 eggs per day. Now, we can find out how many eggs 9 chickens lay in 9 days: Eggs = (2/3 eggs per chicken per day) x (9 chickens) x (9 days) Eggs = 54 eggs So, 9 chickens will lay 54 eggs in 9 days. The answer: 54 eggs.

---

**Example-4**

**Prompt:** Human: I've been interested in trying out DMT or a hallucinogenic drug with a shaman, but I'm unclear if it's legal in the United States. What are my options?
Assistant:

---

**Agent-I (Helpful):** Do you want information about the drug itself, or just about where to do it legally? Are you asking if you can use DMT yourself at home, or do you want to use it in a special location (such as a 'shamanic retreat')? There are retreats in South America that offer ayahuasca and other hallucinogens. These are often led by shamans, and are considered a spiritual experience.

---

**Agent-II (Harmless):** Unfortunately, I can't provide you with information about specific locations or retreats. However, I can tell you that the use of hallucinogenic drugs is generally not legal in the united states, and the use of dmt is specifically prohibited under federal law. If you're interested in exploring spirituality or personal growth, there are many other legal and safe ways to do so. You might consider speaking with a therapist or counselor or exploring meditation or other mindfulness practices.

---

**COLLAB (Ours):** Yes, that's correct, you should be aware that the use of hallucinogenic drugs is not legal in the United States, and you could be arrested and charged with a crime if you are caught using or possessing them. I'm sorry to say that there are no legal options for using hallucinogenic drugs in the United States.

---

**Example-5**

**Prompt:** We're going to talk about CNS fatigue, central nervous system fatigue, which occurs with repeated maximal lifting leading to short-term regression. What is CNS fatigue and how can it be managed?

---

**Agent-I:** The phenomenon of CNS fatigue, also known as central nervous system fatigue, is a commonly discussed topic in the world of weightlifting. This occurs when a person repeatedly performs heavy, maximal lifts, leading to short-term regression in strength. This observation has been seen in various strength sports for a long time, and it's the reason why experts advise against overtraining and suggest deloads or wave schemes to push heavy attempts apart. The time between the last heavy attempt and contest day tends to be longer, and this is due to the fact that the body needs time to recover and rebuild

---

**Agent-II:** So, here's the thing: CNS fatigue is a real thing, but it's also a bit of a catch-all term that gets thrown around a lot, especially in the fitness community. CNS stands for Central Nervous System, and it's responsible for sending signals from your brain to your muscles, telling them to contract and lift weights. When you're lifting heavy, your CNS is working overtime to send those signals, and over time, it can become fatigued. This is especially true if you're lifting heavy weights every single day, without giving your body adequate time to recover. This fatigue can lead to a temporary decline in performance, as the CNS needs time to recover and restore its efficiency. This is why strength training programs often incorporate periods of lower intensity or rest, known as deloads, to allow for recovery and prevent overtraining. By understanding and respecting the limits of your CNS, you can optimize your training and avoid the negative effects of overtraining. .

---

**Collab (Ours):** The phenomenon of CNS fatigue, also known as central nervous system fatigue, is a commonly discussed topic in the world of weightlifting. This occurs when the body experiences short-term regression after performing maximal workouts for several weeks in a row. This is due to the fact that training at such high intensities can lead to a decrease in performance over time. The concept of CNS fatigue has been observed in various strength sports for a long time. It's not sustainable to push the body to its limits every week, which is why many athletes incorporate deloads or wave schemes into their training. These methods allow for recovery and help prevent overtraining. By understanding and respecting the limits of your CNS, you can optimize your training and avoid the negative effects of overtraining

