# OpenReview forum: "Collab: Controlled Decoding using Mixture of Agents for LLM Alignment"
_ICLR.cc/2025/Conference — ICLR 2025 Poster_

### Official Review · Reviewer_Q9zn · 2024-10-30

**Soundness:** 3
**Presentation:** 1
**Contribution:** 4
**Rating:** 5
**Confidence:** 4

**Summary:**

The paper presents the idea of switching between different LLMs during the decoding process in order to maximize the reward of the generated response. To find which policy should be used at each generation step, they define the problem as KL-constrained RL and sample a token greedily with respect to a Q function (regularized by the KL constraint).

**Strengths:**

- The idea of using a Q function to perform routing between different models is novel. Moreover, doing collaborative decoding to maximize reward is an interesting problem that does not have a lot of literature about it.
- The empirical results seem strong, with the proposed algorithm outperforming the baselines.

**Weaknesses:**

The paper is hard to read and understand; in addition, some important details are not discussed at all. Some examples:
- There is no discussion about how the Q functions are trained, although this is a crucial part of the algorithm.
- The only description of the experiments (models, rewards, etc.) is in the appendix. It is also unclear why the author chose these specific experiments.
- The authors explain their method as an extension of CD, which proposes sampling according to equation 3. However, in practice they sample according to equation 5. This is not the same, as the probability under pi_ref is not taken into consideration in equation 5.
- Line 428 hint that you are doing top-K but Algorithm 1 (line 330) talks about top-p, which one is true?

I believe that the technical work presented in the paper is good, but the writing needs to be improved substantially.

**Questions:**

Control Decoding and other similar works [1] use $Q^{\pi_{ref}}$ to augment the decoding. This is not an approximation of $Q^*$ as mentioned in lines 260-261 of the paper, but an alternative solution to the KL-constrained RL problem, known as RWR [2]. The advantage of using $Q^{\pi_{ref}}$ is that it can be easily learned using trajectories from $\pi_{ref}$.
Collab, on the other hand, uses $Q^{\pi_i}$ as an approximation to $Q^*$. Does the author have an explanation (either empirical or theoretical) as to why this is better than just using $Q^{\pi_{ref}}$? This is an important design choice in the algorithm that I feel hasn’t been discussed enough.

In the related work section, I’m missing a discussion about the connections to hierarchical RL. Collab is very close to it, as it learns a policy (parametrized as Q function) to choose which one of several policies to sample from in each state.

I find some of the metrics used in the experiment section are not related to the motivation behind the algorithm. The idea, as presented in the paper, is to use models trained on different reward functions to perform decoding that will maximize a new one. In the experiment section, however, you evaluate non-reward metrics like win rate, diversity, and coherence. Let’s take win rate, for example. Is your claim that the new reward function correlates with the win rate better than the ones that the original models were trained for? And therefore, a higher win rate is equivalent to a higher reward? If so, where is the evidence?

[1] Han, Seungwook, et al. "Value Augmented Sampling for Language Model Alignment and Personalization." arXiv preprint arXiv:2405.06639 (2024).
[2] Peters, Jan, and Stefan Schaal. "Reinforcement learning by reward-weighted regression for operational space control." Proceedings of the 24th international conference on Machine learning. 2007.

---

> ### Author Response · Authors · 2024-11-21
> **Response to Reviewer Q9zn - Part 1**
>
> **General Response**: We thank the reviewer for highlighting and appreciating the novelty of our proposed approach as well as acknowledging that reward maximization through collaborative decoding is an important and underexplored problem.
>
> >Weakness 1: There is no discussion about how the Q functions are trained, although this is a crucial part of the algorithm.
>
> **Response to Weakness 1** Thank you for raising this point. We would like to emphasize that our approach is a completely training-free method. We estimate the Q-function based on its definition in Equation 1 (Eq. 6), using stochastic unbiased samples of the Q-estimate (similar to [1, 2]), and select the token with the maximum implicit Q-value, as outlined in Equation 7.
> That said, it is indeed possible to train an offline Q-adapter (implicit-Q in our case) in a lightweight manner, similar to CD-Fudge [3], to enable faster inference and reduce the time complexity to a constant factor of 𝑘 over naive decoding. We appreciate you highlighting this point and will include a detailed description in the updated draft.
>
> We also want to highlight that the primary focus of this work is to propose a principled method for achieving multi-agent alignment (as also highlighted by the reviewer) using implicit-Q, supported by theoretical guarantees and extensive evaluations.
>
> >Weakness 2: The only description of the experiments (models, rewards, etc.) is in the appendix. It is also unclear why the author chose these specific experiments.
>
> **Response to Weakness 2** :  Thanks for this point. We provide details and motivation for the choice of the experiments below.
>
> **Details of Agents** : Our approach leverages readily available open-source aligned LLMs, fine-tuned for a variety of tasks and datasets. Specifically, we utilize a diverse set of open-source LLMs, including Zephyr: Creative writing and question-answering, Starling: Open-chat and general-purpose dialogue, DolphinQwen: Math word puzzles, cognitive reasoning, and logical reasoning, TalinboyQwen: Creative writing, DolphinMistral: Coding instructions and reasoning.
>
> **New Experiment** : To explicitly highlight the impact of our approach over single-agent decoding (in addition to Fig 4 in paper), we perform an additional experiment (Rebuttal) with two distinct LLM agents: one trained exclusively on the helpfulness subset of HH-RLHF (ChenmieNLP/Zephyr-7B-Beta-Helpful) and the other on the harmlessness subset of HH-RLHF (ChenmieNLP/Zephyr-7B-Beta-Harmless) to generate text that is both helpful and harmless. The results clearly demonstrate that our proposed decoding strategy provides a significant boost in the quality of the generated text, as indicated by higher rewards, compared to single-agent decoding
>
> | Method            | Normalized Avg Reward |
> |------------------------|------------|
> | BoN                   |    0.41    |
> | Agent-1 (Helpfulness)    | 0.3       |
> | Agent-2 (Harmlessness)    | 0.19    |
> | Collab (Ours)         | 1.0    |
>
> **Motivation**: To underscore the generality of our approach, ensure the reproducibility of results, and mitigate potential training biases, we exclusively leveraged fully open-source, off-the-shelf models and datasets. However, we have additional experiments requested by Reviewers and keep on updating with additional experiments with more diverse agents and tasks. We will also move the experimental details and descriptions in the main body as suggested by the reviewer for better clarity.

---

> > ### Comment · Reviewer_Q9zn · 2024-11-23
> >
> > I thank the authors for their response. However, I'm still confused. You mention that:
> > > We estimate the Q-function based on its definition in Equation 1 (Eq. 6), using stochastic unbiased samples of the Q-estimate (similar to [1, 2])
> >
> > However, papers 1 (CD) and 2 (TQ*) perform this estimation in very different ways. CD trains a value function using either FUDGE-style loss or Q learning loss, while TQ* uses an aligned LLM (and its reward model) to obtain an estimation. Which one of the two are you doing?
> >
> > If it is CD, please provide more information on the training process. If it is TQ*, please provide information on which LLM and reward model you used for estimating the Q (and on the sampling process involved in estimating the Q- do you sample separately for every $\pi$? or multiple times?).

---

> ### Author Response · Authors · 2024-11-21
> **Response to Reviewer Q9zn - Part 2**
>
> >Weakness 3: The authors explain their method as an extension of CD, which proposes sampling according to equation 3. However, in practice, they sample according to equation 5. This is not the same, as the probability under pi_ref is not taken into consideration in equation 5.
>
> **Response to Weakness 3** : Thanks for this catch. There is a typo in Eq.5, which will be $Q^*$ (not $Q^{\pi}$) w.r.t to the target reward function (in-line with CD[1] and TransferQ*[2]) where our key objective is to best estimate the $Q^*$ with implicit-Q function leveraging multiple-agents, as shown in Theorem-1, where we quantify the gap w.r.t the true $Q^*$ .
>
> >Question 1 : Control Decoding and other similar works [1] use $Q^{\pi_{\text{ref}}}$ to augment the decoding. This is not an approximation of  $Q^*$ as mentioned in lines 260-261 of the paper, but an alternative solution to the KL-constrained RL problem, known as RWR cite{2}. The advantage of using $Q^{\pi_{\text{ref}}}$ is that it can be easily learned using trajectories from $\pi_{\text{ref}}$. Collab, on the other hand, uses $Q^{\pi}$ as an approximation to $Q^*$. Does the author have an explanation (either empirical or theoretical) as to why this is better than just using $Q^{\pi_{\text{ref}}}$? This is an important design choice in the algorithm that I feel that I feel hasn’t been discussed enough.
>
> **Response to Question 1** Thanks for this question! Below, we provide a comprehensive explanation:
>
> First, we highlight that if we have access to the true $Q^*$ w.r.t to the $r_{\text{target}}$, then one can directly leverage the closed form of the optimal policy in Equation 1, due to the strong convexity of the KL regularized problem. However, $Q^*(s_t,z)$ is never available in practice, hence CD[1] leverages samples from $\pi_{\text{ref}}$ to estimate $Q^*$ whereas TransferQ*[3] relies on an aligned model along with the corresponding reward function to estimate $Q^*$ using their indirect transfer method. In this paper, we take a different route where we leverage multiple off-the-shelf LLMs to best estimate Q-star without any information of individual reward functions (on which these off-the-shelf LLMs are aligned to). Specifically for selecting the next action, we compute the Q-value for each agent and token with respect to $r_{\text{target}}$ and select the token (corr. agent) with the highest implicit Q-value as shown in Eq 7, providing a much better estimate of $Q^*$ under the given conditions.
>
> We agree with the reviewer that it is our design choice. However, it is important to note that in Theorem 1, we estimate the sub-optimality w.r.t the true $Q^*$ as shown in equation as $\Delta = Q^{\pi^*}\_\{\text{target}}(s\_t,z) - Q^{\pi\_{\text{alg}}}\_{\text{target}}(s\_t,z) $ and we upper-bound this term w.r.t $min\_j \delta\_{*j}$ which indicates that our design policy by maximizing the implicit Q-function, will always do equally or better than the best policy in the policy set, thereby justifying our design choice mathematically.
> We agree with the reviewer and will add a detailed justification of the design choice connecting with the theoretical results in the main paper.
>
> > Question 2: In the related work section, I’m missing a discussion about the connections to hierarchical RL. Collab is very close to it, as it learns a policy (parametrized as Q function) to choose which one of several policies to sample from in each state.
>
> **Response to Question 2** Thanks for this very insightful comment and we acknowledge that its  a very interesting connection. We agree that HRL is a general framework and Collab can indeed be formulated as a special case for HRL, where the upper agent provides the goal (reward/Q). We will definitely add this very interesting connection to HRL as a potential scope for future research in the final draft.
> However, since this is one of the first works to provide a principled method of mixture of agent based decoding, we focused on establishing a robust theoretical framework and conducting empirical evaluations to demonstrate the effectiveness of our approach​

---

> ### Author Response · Authors · 2024-11-21
> **Response to Reviewer Q9zn - Part 3**
>
> >Weakness : Line 428 hint that you are doing top-K but Algorithm 1 (line 330) talks about top-p, which one is true?
>
> **Response to Weakness** Thanks for pointing this typo, it will be top-p only. Corrected in the updated draft.
>
>
> > Question 3 : I find some of the metrics used in the experiment section are not related to the motivation behind the algorithm. The idea, as presented in the paper, is to use models trained on different reward functions to perform decoding that will maximize a new one. In the experiment section, however, you evaluate non-reward metrics like win rate, diversity, and coherence. Let’s take win rate, for example. Is your claim that the new reward function correlates with the win rate better than the ones that the original models were trained for? And therefore, a higher win rate is equivalent to a higher reward? If so, where is the evidence?
>
> **Response to Question 3**  Thank you for raising this point. We would like to emphasize that we have thoroughly evaluated and compared our algorithm against the baselines in terms of average target reward, as illustrated in Figure 2 (Evaluations 1–8) and Figure 4. These results demonstrate the superior performance of our algorithm over the baselines. [We believe the reviewer might have missed it due to its positioning along with Table-1 Win-rate evaluations].
>
> In addition, we conducted GPT-4 win-rate-based comparisons with the baselines, which is a widely accepted standard in alignment research [1, 2, 3]. This approach is motivated by the fact that reward models are often biased toward spurious patterns (such as response length), whereas GPT-4, when appropriately prompted, offers a more unbiased, efficient, and fair evaluation framework.
> Furthermore, we assessed our method on coherence and diversity metrics, as outlined in ARGS [3] and Transfer Q* [2]. Since our approach involves token switching, ensuring coherence is critical to maintaining the logical flow of responses. Diversity metrics, on the other hand, evaluate the breadth and novelty of the generated outputs, reflecting the robustness of our method.
> In summary, we conducted comprehensive evaluations using both reward-based and non-reward-based metrics. Our results consistently show that our algorithm outperforms the baselines across all metrics, reaffirming its effectiveness. We have also performed additional experiments on more complex benchmarks like Alpaca-farm, and HH (rebuttal) and shown the improvements of our algorithm in average reward values as well.
>
>
> [1] Mudgal, S., Lee, J., Ganapathy, H., Li, Y., Wang, T., Huang, Y., Chen, Z., Cheng, H.T., Collins, M., Strohman, T. and Chen, J., 2023. Controlled decoding from language models. arXiv preprint arXiv:2310.17022.
>
> [2] Chakraborty, S., Ghosal, S.S., Yin, M., Manocha, D., Wang, M., Bedi, A.S. and Huang, F., 2024. Transfer Q Star: Principled Decoding for LLM Alignment. arXiv preprint arXiv:2405.20495.
>
> [3] Khanov, M., Burapacheep, J. and Li, Y., 2024. ARGS: Alignment as reward-guided search. arXiv preprint arXiv:2402.01694.

---

> ### Author Response · Authors · 2024-11-23
> **Response to Reviewer Q9zn**
>
> We thank the reviewer for appreciating our response and providing us with an opportunity for further clarification.
>
> > Question: However, papers 1 (CD) and 2 (TQ*) perform this estimation in very different ways. CD trains a value function using either FUDGE-style loss or Q learning loss, while TQ* uses an aligned LLM (and its reward model) to obtain an estimation. Which one of the two are you doing?.....
>
> **Response to Question 1**:  Thanks for this question. In our algorithm, we estimate each policy's Q-function (implicit) with stochastic samples of the trajectory from the corresponding policy (ref eq 6). Specifically, for each token, we estimate the implicit Q-function of individual policy and then select the token (from the agent) with the maximum Q-value as shown in Equation 7, where there is a max over the agents (j) and the tokens (z). Q-function of the specific policy/agent (j) and the token (z) is estimated by sampling the trajectory and then evaluating the trajectory with reward $r\_{\text{target}}$. This design choice allows us to achieve tight sub-optimality bounds w.r.t the true/optimal Q* for the reward function $r\_{\text{target}}$ as shown in Theorem 1. Hope this clears the confusion.
>
> Note:  We want to highlight that although the practical estimation approach of the Q-function is different in CD, TQ^* as the reviewer correctly mentioned. However, the basic definition and notion of Q-value are similar in both papers for example Eq-1 (value function in CD) Eq-2 in TQ^*, and can be estimated with stochastic unbiased sampling of the trajectory and evaluation with the reward.
>
> We fully agree that estimating the Q-function on the fly adds computational overhead to inference time, which increases with the number of agents. However, in this work, we primarily focussed on developing a principled framework for multiagent alignment via decoding with theoretical guarantees and empirical evaluations. That said, it is indeed possible to train an offline Q-adapter (implicit-Q in our case) in a lightweight manner, similar to CD-Fudge, to enable faster inference and reduce the time complexity.
>
> Hope we are able to address the reviewer's confusion and are happy to provide any additional clarifications if needed.

---

> > ### Comment · Reviewer_Q9zn · 2024-11-23
> >
> > Thank you for the explanation. I now fully understand the proposed algorithm. However, I still hold the opinion that the paper, in its current state, does not meet the standard for acceptance. My concerns are as follows:
> > 1. **Clarity and Reproducibility**: The paper is written in a confusing manner, making it difficult to understand the details and reproduce the results. Furthermore, both the original and current versions contain mistakes. For instance, the authors claim to use top-p decoding in Algorithm 1, but as indicated on line 428, it appears they actually mean top-k decoding (see: [Top-p sampling - Wikipedia](https://en.wikipedia.org/wiki/Top-p_sampling)).
> > 2. **Computational Feasibility**: The computational cost of the proposed algorithm is not just high—it is extraordinarily demanding. At each decoding step, the algorithm requires generating an entire trajectory from each policy for every top-p token. For example, in the experiments presented in the paper, 10 tokens per policy are used. **This means that 20 full trajectories must be generated just to decode a single token**. This raises serious concerns about the practicality of the method and questions its comparative performance against other approaches, such as BoN, when operating under equivalent computational or inference time constraints.

---

> ### Author Response · Authors · 2024-11-24
> **Response to Reviewer Q9zn**
>
> >Question: Clarity and Reproducibility: The paper is written in a confusing manner, making it difficult to understand the details and reproduce the results. Furthermore, both the original and current versions contain mistakes. For instance, the authors claim to use top-p decoding in Algorithm 1, but as indicated on line 428, it appears they actually mean top-k decoding (see: Top-p sampling - Wikipedia).
>
> **Response**  We apologize for the confusion in terminology. We meant sampling the top-K tokens (not nucleus sampling) and used $p$ instead of $K$ just as a variable to avoid overlap with the number of agents (denoted as $K$) inadvertently causing this misunderstanding. To clarify, we will update the draft to use $M$ for the number of agents and correctly refer to top-K sampling. However, we emphasize that this was a minor confusion in terminology and *not a technical issue or concern of our work*.
>
>
> > Question: Computational Feasibility: The computational cost of the proposed algorithm is not just high—it is extraordinarily demanding. At each decoding step, the algorithm generates an entire trajectory from each policy for every top-p token. For example, in the experiments presented in the paper, 10 tokens per policy are used. This means that 20 full trajectories must be generated just to decode a single token. This raises serious concerns about the practicality of the method and questions its comparative performance against other approaches, such as BoN, when operating under equivalent computational or inference time constraints.
>
> **Response **: Thank you for this question and for providing an opportunity for detailed clarifications. However we believe there is slight confusion regarding the key contributions of our work,
>
> **Key Contribution of the work**:  First we begin by highlighting that the key contribution of our work is to develop a principled approach of combining multiple off-the-shelf LLM optimally to general responses which maximize the target reward function, which was missing from existing literature.  Our work thus not only formulated the problem in a principled manner but also developed the first algorithm with theoretical guarantees showing an optimal way of combining multiple aligned LLMs and the experiments were provided to serve as a proof of concept to demonstrate the practical optimality of our algorithm. Providing a computationally tractable mixture of agent algorithms is important but not the key focus or contribution of our work.
>
> **Empirical Performance & Time Comparison**: We agree with the reviewer that estimating the Q-function on the fly adds computational overhead to inference time, which scales with the number of agents. However, for Collab with 2 agents, we report an average inference time of 68 seconds per prompt (with efficient caching inspired by TQ*, ARGS) compared to 38 seconds for TQ* and 12 seconds for BoN sampling. This slight increase in latency is justified by the 2x improvement in average reward achieved by Collab (Figures 2, 4), which is consistent in all our experimental results.
>
> | Algorithm              | Inference Time | Avg Reward |
> |------------------------|----------------|------------|
> | BoN Sampling           | 12s             | 0.12      |
> | CD/TQ* (Single Agent)  | 38s            | 0.45       |
> | Collab (Multiagent)    | 68s            | 1.0        |
>
> To further improve computational traceability, one can train an offline Q-adapter (function-approximation) in a lightweight manner, similar to CD-Fudge, to enable faster inference (8s per prompt) and reduce the time complexity and **is not a bottleneck**.
>
> **Summary**: We want to remark the key contribution of our work lies in providing a principled method for combining multiple LLMs with provable guarantees in an optimal way, supported by empirical demonstrations to validate the optimality—one of the first works in this direction. Therefore, we feel it is not entirely fair to evaluate the contribution of our work based solely on the computational traceability of Q-function estimation, as it is not the primary focus of our research.

---

> ### Comment · Area_Chair_raYG · 2024-11-28
> **How do rewards for baselines scale as we increase their inference time?**
>
> Ideally, an apples-to-apples comparison with BoN and CD/TQ* so that they reach 68s for would be great to see how the proposed method scales with inference compute.

---

> > ### Author Response · Authors · 2024-12-02
> > **Response to Area Chair raYG**
> >
> > Dear AC,
> >
> > Thanks a lot for the important question and for your interest in the proposed approach. We provide a response to your point below.
> >
> > **Experimental Evidence**:  We conducted additional evaluations to match the inference time of Collab with TQ* with an improved estimation of Q-star.  Specifically, we increased the trajectory length for estimating the Q-star such that the inference latency scaled to approximately 70 seconds for both agent decoding.  For this evaluation, we sampled 500 prompts from the Berkeley Nectar dataset and used Evaluation-4 (see Table 3 in Appendix): Dolphin-2.6-Mistral-7B-DPO as LLM-1, Starling-7B-$\alpha$ as LLM-2, and Mistral-7B-$\alpha$-IT as the reward model.
> >
> >
> > |                Method        | Avg. Normalized Reward | Inference-Time|
> > |------------------------|------------------------|---------|
> > | TQ* (Agent-I) |                 0.45       | 70 sec
> > | TQ* (Agent-II)  |                   0.18     | 70 sec
> > | Collab     |                   1.0     | 68 sec
> >
> > Further, in the same setup, we observed that Collab achieved a win rate of **58.19%** and **64.58%** against Agent-I and Agent-II, respectively. The results highlight that even via scaling inference time, Collab outperforms single-agent decoding methods, which our theory also suggests (as shown in Theorem 1).
> >
> > We will include additional results for other benchmarks in our updated draft, by increasing the inference latency for baselines.
> >
> >
> > Regards
> > Authors

---

> ### Comment · Area_Chair_raYG · 2024-12-04
> **Clarification about results and Missing BoN comparison**
>
> Previously you posted that TQ* gets 0.45 with an inference time of 38s and now it seems increasing inference time to 70s doesn't improve the results. Is that correct?
>
> Also, BoN is a much simpler and widely used baseline -- so it is probably quite important to compare to it.

---

> ### Author Response · Authors · 2024-12-04
> **Further Clarification on Scaled Inference time comparison**
>
> Dear AC,
>
> Thank you for your insightful comment and we truly appreciate the depth of your engagement. We take this opportunity to provide a detailed response as follows:
>
> **Theoretical Justification:**
> **Our benefit comes from the proposed optimal collaboration:** We would like to emphasize that our results align with the theoretical insights presented in our work. We note that just increasing inference time does not yield significant improvement because the performance of single-agent algorithms is fundamentally constrained by their reward difference relative to the target reward. In contrast, our proposed approach performs better because it optimally combines multiple agents, as evidenced by the $\min_j \delta_{ij}$ term in Theorem 1. This highlights the novelty and importance of our contribution: demonstrating a principled approach for combining LLMs to generate optimal responses.
>
> > Previously you posted that TQ* gets 0.45 with an inference time of 38s and now it seems increasing inference time to 70s doesn't improve the results. Is that correct?
>
> **Response** : Thanks for this point. However, we want to point out a minor clarification. Specifically, we wanted to clarify that the previous results, where TQ* obtained an average reward of 0.45, were based on the setup mentioned in: *Evaluation 1:* Dataset: Berkeley Nectar; Agent-I: Zephyr-7B-$\alpha$; Agent-II: Starling-7B-$\alpha$; Reward Model: Mistral-7B-$\alpha$-IT
>
> For the new results with increased latency, we used the setup in: *Evaluation 4:* Dataset: Berkeley Nectar; Agent-I: Dolphin-2.6-Mistral-7B-DPO; Agent-II: Starling-7B-$\alpha$; Reward Model: Mistral-7B-$\alpha$-IT (as mentioned in the previous response).
>
> We observed that increasing the inference latency from 38 seconds to 70 seconds indeed slightly improves the TQ* average reward for both agents. For clarification, we have posted both the original results and the ones with increased latency.
>
> |                Method        | Avg. Normalized Reward | Inference-Time|
> |------------------------|------------------------|---------|
> | TQ* (Agent-I) |                 0.38       | 38 sec
> | TQ* (Agent-II)  |                   0.09     | 38 sec
> | Collab     |                   1.0     | 68 sec
>
> |                Method        | Avg. Normalized Reward | Inference-Time|
> |------------------------|------------------------|---------|
> | TQ* (Agent-I) |                 0.45       | 70 sec
> | TQ* (Agent-II)  |                   0.18     | 70 sec
> | Collab     |                   1.0     | 68 sec
>
>
> **Remark**: We initially compared our approach with the SoTA transfer decoding method (TQ*) since BoN sampling performance was similar to Agent-1 (Figure 2, Evaluation 4). However, we completely acknowledge and agree that BoN is an extremely crucial baseline and simple to scale. As per your suggestion, we are now incorporating BoN sampling with increased test time compute and will add them in the updated draft.
> We thank the AC for this very insightful question and for taking such a deep interest in our work.
>
>
> Regards
>
> Authors

---

### Official Review · Reviewer_4LCR · 2024-11-04

**Soundness:** 3
**Presentation:** 3
**Contribution:** 3
**Rating:** 6
**Confidence:** 4

**Summary:**

The paper proposes a method to elaborate multiple LLMs at the time of inference to combine the strengths of the models. For each token generation, the most promising policy to generate the rest of the sequence is selected. The method is evaluated on two generic alignment tasks using multiple generic LLMs, outperforming single-LLM decoding algorithms.

**Strengths:**

- How to elaborate multiple LLMs is an important research question. Given that we have little knowledge on how to investigate the strengths and weaknesses of the LLMs (as of now), ensemble methods should be useful.
- The method is simple and intuitive.
- The theoretical result is nice to have, yet its practical implication seems not immediate to me.

**Weaknesses:**

> (p.10 l.536) Empirical evaluations demonstrate its superiority over traditional single-agent decoding baselines, providing a robust and computationally efficient method for model alignment in complex, real-world scenarios

- I failed to understand the computational cost of the method. My understanding is that it requires the inference of the whole sequence for each policy, per each token. So the computational cost of generation a sequence of length N would be O(N^2) times of query to LLMs. It would be nice to have a walltime of the algorithm compared with BoN sampling and Agent-1 and 2.

- Although the strength of the method is claimed to be able to adapt to a diverse set of tasks, the experiments are not designed to evaluate in such a scenario. It would be beneficial to evaluate the method for each subtask rather than showing the aggregated result. For example, how does the model perform in the Harmlessness and Helpfulness subsets in HH-RLHF datasets? What kinds of tasks benefit from the ensemble? It would be helpful if we have a post-hoc analysis of what makes the proposed method better than a single-LLM method in the empirical scenario.

**Questions:**

- What is the computational complexity and the walltime of the algorithm in the experiment? I understand that the walltime depends on the hardware and the system but would be a good reference to understand the effectiveness.

- Which algorithm does the SoTA decoding strategy refer to? Is it Transfer Q* (Souradip Chakraborty et a l. 2024)? Or is it controlled decoding (Sidharth Mudgal et al. 2024), which is called SoTA in Chakraborty et al.? A paper should use a term that uniquely identifies the subject rather than a term that changes over time.

- (p.7 Example) Why can COLLAB correctly answer the question when both Agent-1 and 2 fail to answer the question? Is there a reason COLLAB decoding can acquire an ability neither of the Agents have?

---

> ### Author Response · Authors · 2024-11-21
> **Response to Reviewer 4LCR - Part 1**
>
> **General response** We thank the reviewer for appreciating the novel and critical contributions of our work and for acknowledging the simplicity and intuitiveness of our approach.
>
> >Weakness 1 : I failed to understand the computational cost of the method. My understanding is that it requires the inference of the whole sequence for each poli.......It would be nice to have a walltime of the algorithm compared with BoN sampling and Agent-1 and 2.
>
> >Question 1 : What is the computational complexity and the walltime of the algorithm in the experiment? I understand that the walltime depends on the hardware and the system but would be a good reference to understand the effectiveness.
>
> **Response to Weakness 1:** Thank you for your question. We report the inference time required to generate a response for a single prompt in the table. To account for variability in prompt lengths, the inference-time has been averaged over 100 prompts. We observe that, Collab (2 agents) takes 68 seconds to generate a response for a prompt, compared to 38 seconds for TQ* and 8 seconds for naive decoding. However, this slight increase in inference latency is justified by the 2x improvement in average reward by Collab.
>
> | Algorithm              | Inference Time | Avg Reward |
> |------------------------|----------------|------------|
> | Naive Decoding         | 8s             | 0.23       |
> | BoN Sampling           | 12s             | 0.12      |
> | CD/TQ* (Single Agent)  | 38s            | 0.45       |
> | Collab (Multiagent)    | 68s            | 1.0        |
>
> We agree with the reviewer and acknowledge that as the number of agents increases, the proposed multi-agent decoding approach does introduce additional computational overhead at inference time. However, we emphasize that the primary focus and contribution of this work is to provide a principled approach for achieving multi-agent alignment through decoding, supported by theoretical guarantees and empirical evaluations.  However, for efficient implementation, similar to CD-FUDGE method in [A], one can train a small Q-function (implicit Q in our case) adaptor offline, which would allow for faster inference time. This significantly reduces time complexity, similar to ARGS [B], which only introduces a constant factor of p (top-p tokens) over classical decoding methods.
>
>
> >Weakness 2 : Although the strength of the method is claimed to be able to adapt to a diverse set of tasks, the experiments are not designed to.....e model performs in the Harmlessness and Helpfulness subsets in HH-RLHF datasets? ... helpful if we have a post-hoc analysis of what makes the proposed method better than a single-LLM method in the empirical scenario.
>
> | Method            | Normalized Avg Reward |
> |------------------------|------------|
> | BoN                   |    0.41    |
> | Agent-1 (Helpfulness)    | 0.3       |
> | Agent-2 (Harmlessness)    | 0.19    |
> | Collab (Ours)         | 1.0    |
>
> **Response to Weakness2**: Thank you for your suggestion. As recommended by the reviewer, we conducted additional evaluations on the HH-RLHF dataset. To explicitly highlight the importance of mixture-of-agents decoding, we considered two distinct LLM agents: one trained exclusively on the helpfulness subset of HH-RLHF (ChenmieNLP/Zephyr-7B-Beta-Helpful) and the other on the harmlessness subset of HH-RLHF (ChenmieNLP/Zephyr-7B-Beta-Harmless). The objective was to generate text that is both helpful and harmless. To achieve this, we utilized a reward model (Ray2333/reward-model-Mistral-7B-instruct-Unified-Feedback) trained on both subsets of the HH-RLHF dataset. The normalized average rewards (normalized as described in Appendix E.1) for responses generated from 300 prompts in the HH-RLHF dataset, across various decoding strategies, are reported in the table above.
> The result clearly demonstrates that our proposed decoding strategy significantly boosts the quality of the generated text, as indicated by higher rewards, compared to single-agent decoding. This also underscores the importance of token-level switching for tasks that involve balancing two distinct/diverse preferences, such as helpfulness and harmlessness, simultaneously, which is difficult for individual agents.
>
> **Post-hoc analysis** -  Furthermore, as suggested by the reviewer, we have included a post-hoc analysis in Appendix G (Examples 3, 4, 5), providing a qualitative comparison of the text generated by single agents and Collab for various tasks in the updated draft. Agent-I, trained on the helpful subset, risks generating potentially harmful responses by providing information that could encourage illegal activity. Conversely, Agent-II, focused on harmlessness, avoids addressing the user's query altogether, offering only general suggestions for alternative activities. In contrast, our model (Collab) integrates both perspectives effectively, ensuring the response is both helpful in clarifying the legal status and harmless by discouraging illegal behavior.

---

> ### Author Response · Authors · 2024-11-21
> **Response to Reviewer 4LCR - Part 2**
>
> >Question 2 : Which algorithm does the SoTA decoding strategy refer to? Is it Transfer Q* (Souradip Chakraborty et a l. 2024)? Or is it controlled decoding (Sidharth Mudgal et al. 2024), which is called SoTA in Chakraborty et al.? A paper should use a term that uniquely identifies the subject rather than a term that changes over time.
>
> **Response to Question 2**  We refer to Transfer Q* [C] as the SoTA for single-agent decoding (as mentioned in Line 503). We will make it explicit in the updated version to avoid confusion, thanks for the point.
>
> >Question 3 : (p.7 Example) Why can COLLAB correctly answer the question when both Agent-1 and 2 fail to answer the question? Is there a reason COLLAB decoding can acquire an ability neither of the Agents have?
>
> **Response to Question 3** This is an excellent question, and we thank the reviewer for identifying this subtle point. We agree and would like to begin by emphasizing that our theoretical analysis (Theorem 1) establishes that Collab will not perform worse than the best model in the policy set, which is consistently observed across all our experiments. However, in this specific example, Collab successfully produces the correct response even when both individual agents fail. This observation does not contradict our theoretical results; rather, it complements them by showcasing the additional strengths of Collab in leveraging the collective potential of multiple agents to achieve outcomes beyond the capabilities of individual agents.
>
> Practically, we believe that the mixture-of-agents token selection initiates a new trajectory after a few tokens, forming a unique trajectory that neither individual agent would produce on its own, ultimately leading to the correct answer. More specifically, if $x_1,y_1, z_1$ are the tokens generated by Agent-1, Agent-2, and Agent-3 up to a certain point (i.e $x_1$ is the chosen token at t =1 generated by agent 1, $x_2$ is chosen token generated by agent 2 and $x_3$ by agent 3 at t=3), the next token generated by Agent-i will be conditioned on this mixed history of states, i.e $\pi_i(.|x1, y1, z1)$ which has not occurred in individual decoding, thereby creating a new trajectory that ultimately leads to a novel and potentially correct response.
>
> As this work is among the first to explore a principled mixture of agents approach, we plan to further investigate these empirical benefits and provide a more comprehensive theoretical justification in future studies.  We will add the detailed discussions in our final draft which will improve the clarity of the proposed method and we sincerely thank the reviewer for highlighting these critical points.
>
> [A] Mudgal, S., Lee, J., Ganapathy, H., Li, Y., Wang, T., Huang, Y., Chen, Z., Cheng, H.T., Collins, M., Strohman, T. and Chen, J., 2023. Controlled decoding from language models. arXiv preprint arXiv:2310.17022.
>
> [B] Khanov, M., Burapacheep, J. and Li, Y., 2024. ARGS: Alignment as reward-guided search. arXiv preprint arXiv:2402.01694.
>
> [C] Chakraborty, S., Ghosal, S.S., Yin, M., Manocha, D., Wang, M., Bedi, A.S. and Huang, F., 2024. Transfer Q Star: Principled Decoding for LLM Alignment. arXiv preprint arXiv:2405.20495.

---

> > ### Author Response · Authors · 2024-11-25
> >
> > Dear Reviewer,
> >
> > As the deadline approaches, we wanted to humbly reach out to inquire if there are any remaining concerns or questions. We are more than happy to engage in further discussions and provide clarifications as needed.
> >
> > Regards,
> > Authors

---

> > ### Comment · Reviewer_4LCR · 2024-11-25
> >
> > Thank you very much for the detailed response. Now I think I understand the position of the paper much better.
> >
> > > Inference Time
> >
> > I think it is totally fine to be slower than the others, but it should be reported in the paper (maybe in Appendix).
> >
> > > additional evaluations on the HH-RLHF dataset
> >
> > Thank you very much for the effort. I think the additional experiment supports the claim that the method is also empirically effective.
> >
> > > Why can COLLAB correctly answer the question when both Agent-1 and 2 fail to answer the question?
> >
> > Thank you very much for the additional experiments and for sharing your thoughts.
> > Now I see the benefit of the token-level mixture-of-agent approach.
> >
> > In my stylistic preference, I would like to have your thoughts on the potential of the proposed approach written in the paper (maybe in Dicussion Section), even if you have yet to come up with the theoretical justification.

---

> > > ### Comment · Reviewer_4LCR · 2024-11-25
> > >
> > > > Normalized Avg Reward for HH-RLHF
> > >
> > > Thank you very much for the additional experiments on HH-RLHF. It would be nice if you could show the reward scores for the Helpfulness and Harmlessness subset separately rather than aggregated. I would like to see how much loss we get when we run COLLAB compared to the model specifically trained for one of the subsets. I would guess that there will be a trade-off and would like to know how much it costs.

---

> > > ### Author Response · Authors · 2024-11-29
> > > **Response to Reviewer 4LCR**
> > >
> > > We thank the reviewer for acknowledging our rebuttal and we are glad that it helped in improving the clarity of our proposed approach.
> > >
> > > > additional evaluations on the HH-RLHF dataset
> > >
> > > We acknowledge that the specific experiment on HH-RLHF dataset helped in explicitly highlighting the superiority of our mixture of agents-based approach over baselines.
> > >
> > > > I think it is totally fine to be slower than the others, but it should be reported in the paper (maybe in the Appendix).
> > >
> > > We absolutely agree on this point with the reviewer and will add the wall time and computational details in the updated draft.
> > >
> > > > Why can COLLAB correctly answer the question when both Agent-1 and 2 fail to answer the question?
> > >
> > > As the reviewer correctly highlighted this is a very critical point and we will add a detailed description and limitations in the discussion section of our updated draft.
> > >
> > > **Remark**:  We would like to specifically highlight that the key points raised by the reviewer are extremely insightful and have been very helpful in improving the overall presentation of our work. We thank the reviewer for understanding and appreciating the key contributions of our proposed approach.

---

### Official Review · Reviewer_4fXt · 2024-11-04

**Soundness:** 2
**Presentation:** 3
**Contribution:** 2
**Rating:** 6
**Confidence:** 4

**Summary:**

This paper proposes Controlled Decoding using Mixture of Agents for LLM Alignment, a method that achieves alignment at inference time without retraining the policy model. However, the comparison with classical alignment methods is insufficient and some critical experimental details are unclear.

**Strengths:**

1. The approach of achieving alignment without retraining the policy model is innovative and could significantly reduce computational costs.

2. The concept of using a mixture of agents for decoding provides a fresh perspective on controlling language model outputs.

**Weaknesses:**

1. Is the implicit Q-function an additional trained language model? How does its cost compare to methods based on DPO, PPO, and RLHF?

2. Can the implicit Q-function directly guide a policy model in decoding? Is it necessary to use multiple agents collaboratively for decoding?

3. How were the agents initialized in the experiments? Were these agents explicitly trained for alignment?

4. The experimental section lacks a comparison with alignment algorithms based on DPO and PPO.

5. The authors should consider more complex, objective tasks such as those involving reasoning and math to avoid potential biases from the reward model and GPT-4 evaluations.

**Questions:**

please see weaknesses

---

> ### Author Response · Authors · 2024-11-21
> **Response to Reviewer 4fXt - Part1**
>
> **General Response** We thank the reviewer for recognizing the novelty of our approach in achieving alignment without re-training and for appreciating our use of a mixture of agents for controlled decoding, which offers a fresh perspective.
>
> >Weakness 1 : Is the implicit Q-function an additional trained language model? How does its cost compare to methods based on DPO, PPO, and RLHF?
>
> **Response to Weakness1** Thank you for raising this crucial point. We want to emphasize that our method is entirely training-free, where the Q-function is estimated directly from its definition in Equation 6 by taking stochastic unbiased samples of the Q-value, similar to the approach in [1, 3]. The token with the maximum implicit Q-value is then selected, as shown in Equation 7.
>
> We acknowledge that this approach introduces additional computational overhead during inference - 68s per prompt (2 agents) in comparison with 38s for single-agent decoding methods [1, 3]. However, this overhead is justified by the significant improvement in average reward achieved by Collab, as shown in Figure 2. Additionally, one can train an offline Q-adapter (implicit-Q in our case) to enable faster inference, reducing time complexity to a constant factor of p (top-p tokens) over classical decoding methods [3]. As also highlighted in [1, 2], training a Q-function adapter (CD-Fudge) is lightweight as it involves supervised prediction of a scalar Q-value rather than training a generative policy with DPO/PPO. We provide further details on computational cost below :
>
> **Computational Cost Comparison with RLHF/DPO** : RLHF/DPO involves fine-tuning the model on the target reward and requires significantly more compute than our proposed mixture-of-agent decoding approach. For instance, RLHF training for the Zephyr-7B model requires 6 A6000 GPUs, even with techniques like LoRA and 8-bit quantization, whereas Collab requires only 2 A6000 GPUs for decoding, making it substantially more efficient in terms of compute.
>
> >Weakness 2 : Can the implicit Q-function directly guide a policy model in decoding? Is it necessary to use multiple agents collaboratively for decoding?
>
> **Response to Weakness2**
>
> Yes, the implicit Q-function directly guides the policy model in decoding, as shown in Equation 5, (similar to CD [1], Equation-1) where it updates the probability of the token (under the reference policy) with exponential weighting based on the implicit-Q value. Ideally, if we had access to the true Q-star, Equation 3 would provide the optimal solution to the alignment problem. However, since Q-star is never available in practice, using multiple agents collaboratively helps provide a closer estimate of Q-star through the implicit Q, as demonstrated in Theorem-1, where the sub-optimality gap is measured with respect to the true Q-star. Below, we provide a detailed justification of the use of multiple agents for decoding.
>
>
> **Importance of Multiple Agents in Decoding** : The importance of using multiple agents for decoding can be understood from the upper-bound in Theorem 1, particularly the term $\min_j \delta_{\ast j}$ which represents the minimum difference between the target reward and the reward on which the best policy in the set has been aligned to. If only a single agent is used, the sub-optimality will remain constant and proportional to the reward difference between the target reward and that single agent's reward, which cannot be improved, as shown below :
> $\Delta \leq \delta_{\ast j} + \alpha KL\left(\pi_j(\cdot|s), \pi_{\text{ref}}(\cdot|s)\right) - \alpha KL\left(\pi_{\text{alg}}(\cdot|s), \pi_{\text{ref}}(\cdot|s)\right)$
>
> Consequently, if the single agent ($j^{\text{th}}$) is highly sub-optimal for the target reward i.e $\delta_{\ast j}$ is very high, the performance will suffer significantly. In contrast, leveraging a diverse set of policies can mitigate this gap, as $\min_j \delta_{\ast j}$ can be low for at least one of the policies in the set (even though $\delta_{\ast j}$ is high for some specific agents). This advantage of collaborative multi-agent decoding has been consistently observed in our experimental evaluations.

---

> ### Author Response · Authors · 2024-11-21
> **Response to Reviewer 4fXt - Part2**
>
> >Weakness 3 : How were the agents initialized in the experiments? Were these agents explicitly trained for alignment?
>
> **Response to Weakness 3** Thank you for raising this point. We want to emphasize that our approach leverages already available fully open-source aligned LLMs (Huggingface), fine-tuned for a variety of tasks and open-sourced datasets. Specifically, we utilized a diverse set of open-source LLMs, including Zephyr - Creating Writing, Question-answering, Starling - Open chat, general purpose dialogue, DolphinQwen - Mathword puzzle, Cognitive and logical reasoning, TalinboyQwen- Creative writing, DolphinMistral- Coding instruction, reasoning, etc.
>
> Through our experiments, we aimed to demonstrate that our multi-agent decoding approach is a purely inference-time algorithm capable of leveraging any existing off-the-shelf aligned LLMs. To underscore the generality of our approach, facilitate easy replication of results, and eliminate potential training biases, we exclusively used fully open-source, off-the-shelf models and datasets. Our experimental results demonstrate the benefits of our method, even when using off-the-shelf LLMs. We have also performed additional experiments on challenging tasks- Alpaca farm and HH which shows the efficacy of our approach.
> As suggested by the reviewer, we plan to include additional diverse agents and tackle more challenging tasks in the final version to further strengthen our findings.
>
> >Weakness 4: The experimental section lacks a comparison with alignment algorithms based on DPO and PPO.
>
> Thank you for pointing this out. We have added a comparison with single-agent DPO for Agent-1 and Agent-2 for Evaluation 1 (Table 3 in the paper) and will include additional comparisons (DPO, PPO) for other evaluations in the final version as suggested by the reviewer.
>
> | Algorithms             | Avg Reward (Normalized) |
> |------------------------|------------|
> | BoN                   | 0.09       |
> | Agent-1 (Decoding)    | 0.73       |
> | Agent-2 (Decoding)    | 0.52       |
> | Agent-1 (DPO)         | 0.69       |
> | Agent-2 (DPO)         | 0.41       |
> | Collab (Ours)         | 1.0        |
>
> It is evident, that DPO's performance is slightly less (especially for Agent-2) than single-agent decoding approaches which have also been observed in several recent works [1,2,3,4]. Additionally, we would like to highlight that it has already been shown both theoretically and empirically that optimal inference-time decoding can perform as well as training-based alignment methods (or even improve) in terms of the reward function/win rate (refer to Theorem 1 in Transfer-Q [3]), which aligns with our experimental observations in this work.
>
> >Weakness 5 : The authors should consider more complex, objective tasks .......to avoid potential biases from the reward model and GPT-4 evaluations.
>
> **Response to Weakness 5** Thanks for this suggestion. As requested, we performed ablation on a more complex task - Alpaca Farm as shown below:
>
> | Method            | Avg Reward |
> |------------------------|------------|
> | BoN                   |    25.08    |
> | Agent-1 (Decoding)    | 24.68       |
> | Agent-2 (Decoding)    | 24.29      |
> | Collab (Ours)         | 26.21       |
>
> In the table above, we report the average reward obtained on GPT-4 preference split of Alpaca Farm (tatsu-lab/alpaca_farm) using various decoding strategies on 300 prompts. For this evaluation, we employed Zephyr-7B as Agent-1, Starling-7B as Agent-2, and LLAMA-3-8B as the reward model. The results clearly demonstrate that by utilizing a mixture of agents, Collab achieves a higher average reward compared to other baseline approaches. As per the reviewer's suggestion, we are also conducting evaluations on additional complex tasks such as math reasoning, and will update the final draft with these additional evaluations.
>
>
> References :
> [1]. Controlled Decoding from Language Models
> [2]. ARGS: Alignment as Reward-Guided Search
> [3]. Transfer Q Star: Principled Decoding for LLM Alignment
> [4]. From r to Q-star: Your Language Model is Secretly a Q-Function

---

> > ### Author Response · Authors · 2024-11-25
> >
> > Dear Reviewer,
> >
> > As the deadline approaches, we wanted to humbly reach out to inquire if there are any remaining concerns or questions. We are more than happy to engage in further discussions and provide clarifications as needed.
> >
> > Regards,
> > Authors

---

> > > ### Comment · Reviewer_4fXt · 2024-11-26
> > >
> > > The authors have addressed some of my doubts, so I have increased my score.

---

> > > > ### Author Response · Authors · 2024-11-29
> > > > **Final Response to Reviewer 4fXt**
> > > >
> > > > We thank the reviewer for acknowledging our rebuttal and increasing the score. The discussion with the reviewer has helped to improve the presentation of our proposed approach and we will add the detailed discussion in our updated draft.
> > > >
> > > > Regards
> > > > Authors

---

### Official Review · Reviewer_d8Dt · 2024-11-04

**Soundness:** 3
**Presentation:** 4
**Contribution:** 3
**Rating:** 8
**Confidence:** 3

**Summary:**

This work proposes to extend decoding / inference time alignment using a mixture-of-experts based decoding strategy. For this, the authors make use of existing LLMs as policy experts. The decoding method proposed by the authors uses a token level selection strategy where an LLM is selected from the experts / agents for each token. This selection is performed using a utility metric which is based on the implicit Q function. The authors also provide a theoretical justification of their approach and support it using empirical results.

**Strengths:**

* Dont need to update the billions of parameters that RLHF needs.
* Their approach can make use of specialized models which are experts in a subset of capabilities that are desired.
* Current solutions to mixing multiple experts relies on either tuning a model, or explicit formulas, and rely on expert demonstrations. This work dynamically merges models
* This approach is especially beneficial in settings where reward models / policy parameters of models are not available readily thus impacting a lot of industry applications.
* The theoretical justification and details are sound.
* The paper is generally well written and easy to follow.
* The evaluations are diverse and across 7 different setups, however, only 2 datasets: Berkeley Nectar and HH-RLHF.

**Weaknesses:**

* It is unclear what is the reference policy used in the KL terms that is used to obtain the objective of each policy (J).
* "The sub-optimality gap will be lower when 1) the best agent’s reward function is close to the target reward function, and 2) when the regularization terms are properly controlled so that both the reference policy and the optimal policy are close" -> The choice of the reference policy seems to be important.
* I would have liked to see analysis on compute requirements.
* Minor point: There is no human evaluation done in this work. Evaluating using GPT4 should not be used to claim alignment with humans.
* The main usage of this work seems to be in using domain expert policies that are good in different aspects / tasks. It is unclear how the chosen policies are experts in different areas that are evaluated. For this, I would suggest using evaluation on harder tasks maybe like reasoning / coding datasets (alpaca eval, arena hard, mt bench).

**Questions:**

* What is the distribution of the selection of agents.
* How much is the compute difference between doing RLHF + single decoding vs mixture of decoding?
* Line 97-99 seems repeated.

---

> ### Author Response · Authors · 2024-11-21
> **Response to Reviewer d8Dt**
>
> **General Response** We thank the reviewer for their thoughtful feedback and for recognizing the central contribution of our work—a principled and novel approach to token selection using a mixture-of-experts decoding strategy. We also greatly appreciate the acknowledgment of the theoretical justifications and the clarity of our presentation.
>
> >Weakness 1 & 2 : It is unclear what is the reference policy used in the KL terms that is used to obtain the objective of each policy (J). The sub-optimality gap will be lower when 1) the best agent’s reward function is ..... The choice of the reference policy seems to be important.
>
> **Response to Weakness 1 & 2** Thank you for highlighting this point. Indeed, the reference policy plays a crucial role in the sub-optimality analysis. In our case, the reference policy corresponds to the base policy (pre-trained/reference model) from which the policies $\pi_k, \forall k \in K$, have been fine-tuned or aligned.
>
> >Weakness 3 : I would have liked to see analysis on compute requirements. How much is the compute difference between doing RLHF + single decoding vs mixture of decoding?
>
> Thank you for your question. First, we would like to clarify that RLHF involves fine-tuning the model on the target reward, making it significantly more compute-intensive compared to our proposed mixture-of-decoding approach. Specifically, for the Zephyr-7B model, RLHF training requires 6-7 A6000 GPUs, even with techniques like LoRA and 8-bit quantization. In contrast, Collab requires only 2 A6000 GPUs for decoding (1 A6000 for single agent decoding), making it substantially more efficient in terms of compute requirements. In terms of inference time, Collab takes 68 seconds (2 agents) to generate a response for a prompt, compared to 38 seconds for TQ* and 8 seconds for naive decoding. However, this slight increase in inference latency is justified by the 2x improvement in average reward by Collab (Figure 2, 4) as also highlighted by the reviewer.
>
>
> >Weakness 4: The main usage of this work seems to be in using domain expert policies that are good in different aspects/tasks. It is unclear how the chosen policies are experts in different areas that are evaluated. For this, I would suggest using evaluation on harder tasks maybe like reasoning/coding datasets (alpaca eval, arena hard, mt bench).
>
> **Response to Weakness 4** Thank you for your suggestion. As requested, we performed the ablation on Alpaca Farm shown below:
>
>
> | Method            | Avg Reward |
> |------------------------|------------|
> | BoN                   |    25.08    |
> | Agent-1 (Decoding)    | 24.68       |
> | Agent-2 (Decoding)    | 24.29      |
> | Collab (Ours)         | 26.21       |
>
> In the table above, we report the average reward obtained on GPT-4 preference split of Alpaca Farm (tatsu-lab/alpaca_farm) using various decoding strategies on 300 prompts. For this evaluation, we employed Zephyr-7B as Agent-1, Starling-7B as Agent-2, and LLAMA-3-8B as the reward model. The results clearly demonstrate that by utilizing a mixture of agents, Collab achieves a higher average reward compared to other baseline approaches. As per the reviewer's suggestion, we are also conducting evaluations on additional tasks such as MT-Bench and Arena Hard. We will update the final draft with these additional evaluations.
>
> >Question: What is the distribution of the selection of agents
>
> **Response to Question** In our current experiments, we consider a diverse set of open-source LLMs, (Huggingface) including *Zephyr* (creative writing and question-answering), *Starling* (open chat and general-purpose dialogue), *DolphinQwen* (math word puzzles, cognitive reasoning, and logical QA), *TalinboyQwen* (creative writing), and *DolphinMistral* (coding instructions and reasoning). To underscore the generality of our approach, and eliminate potential training biases, we exclusively used fully open-source, off-the-shelf models and datasets. As suggested by the reviewer, we plan to incorporate more diverse agents and introduce harder tasks in the final version.
>
> We thank the reviewer for the positive feedback and acknowledging the key contributions of our work.

---

> > ### Comment · Reviewer_d8Dt · 2024-11-24
> > **Response to rebuttal**
> >
> > Thank you for clarifying my questions.
> >
> > It would be nice if you can explicitly clarify in the paper about the reference policy. Even after reading it again after you mentioning it is not super clear.
> >
> > I saw the compute requirements and i do believe in it's current state, the latency is too much to make this work usable. It's 2x (as compared to the other best SOTA and 5x compared to BON sampling -- which works generally very well in practice) with just 2 agents. I do feel that this idea can be helpful, maybe in some other way (not necessarily each token?). The main usability of this work is to merge skills of multiple specialized agents that might become prohibitively expensive if you consider 2+ agents. Other reviewers also point out this weakness.
> >
> > Regarding the other results, I'm not sure what the average reward is that you add for Alpaca Farm? How did you select the 300 prompts? You say that you will add more diverse agents and harder tasks in the final version, but can you provide more details?
> >
> > I would like to keep my score same.

---

> > > ### Author Response · Authors · 2024-11-25
> > > **Response to Reviewer d8Dt**
> > >
> > > We are happy that our response could clarify the reviewer's concerns and we deeply appreciate the reviewer's assessment and understanding of the core of work.
> > >
> > > > Point  1: I saw the compute requirements and i do believe in it's current state, the latency is too much to make this work usable. It's 2x ... .oken?)..... might become prohibitively expensive if you consider 2+ agents
> > >
> > > **Response to Point 1**: Thanks for this important point. We agree that performing principled multiagent decoding adds computational overhead to inference time, however, this increase in latency is justified by the 2x improvement in the average reward which is consistent in all our experimental results including the additional results on harder tasks.
> > >
> > > To further improve computational traceability, one can train an offline Q-adapter (function approximation) in a lightweight manner, similar to CD-Fudge, to enable faster inference for Collab (8s per prompt).
> > >
> > > However, we want to highlight that the **key contribution of this work** is to develop a principled approach of combining multiple off-the-shelf LLM optimally to general responses that maximize the target reward function with provable guarantees, which was missing from existing literature.
> > >
> > >
> > > >Point 2: Regarding the other results, I'm not sure what the average reward is that you add for Alpaca Farm? How did you select the 300 prompts? You say that you will add more diverse agents and harder tasks in the final version, but can you provide more details?
> > >
> > > **Reponse to Point 2** :  Thanks for this point. For the Alpaca Farm experiment, we selected 300 prompts randomly from the "GPT-4 preference split" of Alpaca Farm (tatsu-lab/alpaca_farm). We evaluate various decoding strategies using LLAMA-3-8B reward model and report the average reward obtained over the 300 prompts, which clearly shows Collab outperforms existing baselines. We are currently performing additional experiments on  1. other splits on Alpaca farm  2. MT-bench   3. Mathematical reasoning task for GSM-8k and will report the results in the updated draft. We would be happy to incorporate any additional tasks or benchmarks the reviewer may suggest.
> > >
> > >
> > > **Remark**:  We want to highlight that our discussion with the reviewer has been extremely insightful. Several points raised by the reviewer not only helped the key ideas but also improved the overall presentation of our work. We will update the final version of our draft with detailed discussions of the rebuttal.

---

### Meta-Review · Area_Chair_raYG · 2024-12-20

**Metareview:**

The paper introduces a novel controlled decoding strategy employing a mixture of expert LLMs for improved alignment in language models.  This multi-agent approach is claimed to surpass single-agent decoding methods in adapting to diverse tasks and preferences, enhancing test-time performance without retraining.  The findings support this claim, demonstrating COLLAB's superior performance over single-agent baselines and achieving notable improvement in average reward and win-tie rate against GPT-4.

Strengths: The paper presents a principled approach to controlled decoding by leveraging a mixture of expert agents.  It offers a computationally efficient alternative to traditional RLHF methods, allowing for inference-time alignment without retraining.  The theoretical analysis and empirical evaluations are comprehensive, demonstrating the effectiveness of COLLAB in various tasks and preferences.

Weaknesses:  The computational requirements, particularly the inference latency introduced by multi-agent decoding, are not thoroughly analyzed and discussed. More thorough apples-to-apples comparisons in terms of inference budget with baselines (BoN and others), both in terms of inference time as well as number of LLM calls. The absence of human evaluation and the reliance on GPT-4 for alignment claims are also limitations.  Finally, while the paper claims the use of domain expert policies, it remains unclear how the chosen policies demonstrate expertise in the evaluated areas (the authors did run experiments on AlpacaFarm).

Despite the identified weaknesses, some reviewers agree that the paper should be accepted, based on the paper's theoretical grounding, and empirical support. Indeed, COLLAB offers a promising direction for improving LLM alignment and I hope the authors would address the weaknesses in the revision.

**Additional Comments On Reviewer Discussion:**

The rebuttal period focused on clarifying the paper's claims and addressing reviewers' concerns.

- Reviewer d8Dt questioned the clarity of the reference policy and the computational cost.  The authors clarified that the reference policy is the base model from which other policies are fine-tuned, and they discussed the computational efficiency of their approach compared to RLHF.

- Reviewer 4fXt asked about the Q-function, agent initialization, and comparison with other methods.  The authors explained that the Q-function is estimated directly from its definition, the agents are initialized from existing LLMs, and they provided comparisons with DPO.

- Reviewer 4LCR inquired about computational complexity, experimental design, and the SOTA decoding strategy.  The authors reported the inference time, conducted additional evaluations on the HH-RLHF dataset, and clarified the SOTA decoding strategy as Transfer Q*.

- Reviewer Q9zn questioned the training of Q-functions, experimental details, and the sampling method.  The authors clarified that their approach is training-free, provided details about the experiments, and corrected a typo in the sampling method.

- I also requested an apples-to-apples comparison with BoN and TQ* by increasing their inference time to match that of Collab.  The authors conducted additional evaluations, and promised to run BoN experiments. Overall, they partially demonstrated that Collab still outperforms single-agent decoding methods even when the inference time is scaled.

Overall, the authors' responsiveness to the reviewers' concerns and their efforts to improve the paper during the rebuttal period contributed significantly to my decision to accept the paper.

---

### Decision · Program_Chairs · 2025-01-22

Accept (Poster)